# Quantifying persistence in the T-cell signaling network using an optically controllable antigen receptor

Michael J Harris[1],[†] iD, Muna Fuyal[2],[†] iD & John R James[1],[2],[*] iD

## Abstract

T cells discriminate between healthy and infected cells with remarkable sensitivity when mounting an immune response, which is hypothesized to depend on T cells combining stimuli from multiple antigen-presenting cell interactions into a more potent response. To quantify the capacity for T cells to accomplish this, we have developed an antigen receptor that is optically tunable within cell conjugates, providing control over the duration, and intensity of intracellular T-cell signaling. We observe limited persistence within the T-cell intracellular network on disruption of receptor input, with signals dissipating entirely in ~15 min, and directly show sustained proximal receptor signaling is required to maintain gene transcription. T cells thus primarily accumulate the outputs of gene expression rather than integrate discrete intracellular signals. Engineering optical control in a clinically relevant chimeric antigen receptor (CAR), we show that this limited signal persistence can be exploited to increase CAR-T cell activation threefold using pulsatile stimulation. Our results are likely to apply more generally to the signaling dynamics of other cellular networks.

**Keywords** cell signaling; optogenetics; receptors; T cells; transcription factors
**Subject Categories** Biotechnology & Synthetic Biology; Immunology; Signal Transduction
**Mol Syst Biol. (2021) 17: e10091**

## Introduction

The ability of cells to convert extracellular stimuli into information that can guide future decisions is an essential requirement for organisms to survive within their environment (Jordan *et al*, 2000). External inputs are sensed by receptors that reside predominantly at the cell surface in mammalian cells, whereas their functional output is either manifested as changes in gene expression (Pope & Medzhitov,

2018), metabolism (Saxton & Sabatini, 2017), or behaviors such as cell migration (Devreotes & Horwitz, 2015). These outputs can drive long-term alterations in cell function by bringing about a new state that persists when the originating input has been removed and constitutes a memory of previous signaling (Burrill & Silver, 2010). The transduction of signals from the surface receptors to the transcriptome is mediated by an intracellular network of signaling proteins and lipids that transmit and amplify the cellular inputs. While it is well established that signaling networks have significant information processing capabilities (Atay & Skotheim, 2017; De *et al*, 2020; Martin *et al*, 2020), less is known about the persistence of signals within such networks when the initiating stimulus is removed; a quantitative understanding of this response requires elucidating the timescales of how intermediary signals decay within the network. Forward steps in an intracellular signaling network often involve phosphorylation of a protein, which is essentially irreversible through coupling to ATP hydrolysis. These biochemical reactions are invariably countered by an opposing phosphatase activity; the efficiency of these reverse steps will therefore define the level of signal persistence within the network. Any potential for signals to be retained on cessation of receptor input would constitute a short-term memory, which could be observed directly when distinct inputs are separated over time providing a mechanism to combine discrete events into a more potent response.

T-cell activation is a physiological situation where integrating multiple signaling events could drive a more robust output response. T cells are an essential cell type of our adaptive immune system that keep us healthy by continually scanning the surface of other cells looking for signs of infection. The T-cell antigen receptor (TCR) is a multi-protein complex expressed within the plasma membrane that is the primary mediator of this immune surveillance. The TCR can recognize pathogen-derived peptides in the MHC protein complex (pMHC) expressed on almost all cells and the information encoded in this interaction is relayed to the T-cell interior, initiating a signaling cascade that ultimately leads to changes in T-cell activation and cell fate differentiation (Smith-Garvin *et al*, 2009; Brownlie & Zamoyska, 2013). Importantly, T cells are known to be exceptionally sensitive, with the ability to show a detectable

---

1 Molecular Immunity Unit, Department of Medicine, MRC-LMB, University of Cambridge, Cambridge, UK
2 Division of Biomedical Sciences, Warwick Medical School, University of Warwick, Coventry, UK
*Corresponding author. Tel: +44 2765 28367; E-mail: john.james@warwick.ac.uk
†These authors contributed equally to this work

response to cells presenting just a few cognate pMHC ligands (Anikeeva *et al,* 2012; Huang *et al,* 2013).

T cells continuously interact with other cells as they traverse between the blood and lymph circulation systems, primarily engaging with antigen-presenting cells (APCs) in lymph nodes and forming mobile synapses, termed kinapses (Dustin, 2008). Previous work *in vitro* has shown that T cells can be activated over multiple APC interactions even though the individual interactions are likely to be transient, lasting < 20 min (Underhill *et al,* 1999; Gunzer *et al,* 2000), results that have been corroborated from intravital imaging in mice (Celli *et al,* 2005; Marangoni *et al,* 2013; Le Borgne *et al,* 2016). One study also measured the interval between sequential T cell/APC interactions to be 1.5–2 h (Gunzer *et al,* 2000), implying a significant temporal window between signaling pulses. However, it has also been found that sustained proximal signaling is required to drive effective T-cell activation (Valitutti *et al,* 1995; Smith-Garvin *et al,* 2009; Au-Yeung *et al,* 2014), which would require T cells to combine sub-optimal signaling events initiated during these transient encounters, known as "Phase-One" (Mempel *et al,* 2004), into more robust activation. Periodic signal disruption has shown that T cells are capable of combining temporally separated stimuli (Faroudi *et al,* 2003; Munitic *et al,* 2005; Clark *et al,* 2011) but may affect the differentiation pathway of the activated T cells when serial transient interactions are formed (Scholer *et al,* 2008).

To investigate this discrepancy between the requirement for sustained signaling and interactions with multiple APC, we sought to quantify the rate at which receptor signaling dissipates within T cells to define the timescales over which the intracellular network can integrate signals from discrete receptor inputs. Exciting new methods to do this have come from the development of optogenetic toolkits that provide the means to control signaling in a non-invasive and reversible manner (Toettcher *et al,* 2013; Bugaj *et al,* 2017). Recent studies have used these approaches to understand the dynamics of signaling within the ERK pathway and how this information is encoded at the transcriptional level (Wilson *et al,* 2017). There have also been two studies that have used optically controllable ligands to investigate whether the TCR can read out the lifetime of the TCR/pMHC interaction. However, these results were achieved with either soluble tetrameric (Yousefi *et al,* 2019) or bilayer-bound (Tischer & Weiner, 2019) ligands, which may not be capable of driving persistent downstream signaling as they remove the physiological context of the T cell–APC interface and other costimulatory signals. Another recent study has demonstrated that the initiation of calcium fluxing in T cells can be controlled *in vivo* using an optogenetic actuator (Bohineust *et al,* 2020). Chemical biology approaches have also been recently used to engineer a light-inducible "switch" into a chimeric antigen receptor (CAR) to control T-cell function (Zhang *et al,* 2021).

We have developed a new antigen receptor that can synchronously initiate signaling within T cell–APC conjugates, by combining optical and chemically controllable inputs with the essential intracellular signaling domains of the TCR itself. This approach provides powerful control over the tempo and duration of intracellular signaling dynamics. We used this tool to quantify the persistence of signals at representative intracellular nodes within the TCR signal transduction network when receptor input is acutely disrupted. Through this approach, we have found that intracellular signals dissipate completely within ~15 min, with mRNA levels providing

the most persistent intermediary state with a half-life of ~25 min. These values provide a temporal window over which T cells can directly integrate TCR signals between multiple APC interactions, rather than simply accumulating the transcriptional output of gene expression from each stimulation. We also show that this limited signal persistence can be exploited to drive more efficient signaling in primary T cells by transplanting optical control to a clinically relevant chimeric antigen receptor.

# Results

## Construction of an optically modulated chimeric antigen receptor in T cells

Our first goal was to design a synthetic antigen receptor in T cells that would provide a stimulus to the intracellular T-cell signaling network that was under light-mediated control. Furthermore, this optically modulated receptor needed to function within the context of the interface between a T cell and APC. This requirement is important as it allows other signaling pathways to function unperturbed and most closely mimics the native signaling environment. The most efficient way to achieve this was by engineering in the ability to physically uncouple extracellular ligand binding from signal transduction across the plasma membrane (Fig 1A). This was accomplished using the LOVTRAP system (Wang *et al,* 2016), where illumination of a LOV2 domain with blue light (400–500 nm) causes the reversible dissociation of an engineered Zdk domain, which only binds the dark state of LOV2. To create our optically controlled chimeric antigen receptor, which we term an "OptoCAR", we fused the light-sensitive LOV2 domain to the intracellular terminus of a synthetic receptor that we have previously shown can replicate the function of the native TCR complex similarly to other CAR structures (James & Vale, 2012; James, 2018). The ITAM signaling motifs from the TCR complex (ζ-chain) were fused to the Zdk domain, and this intracellular part of the OptoCAR was constrained to the plasma membrane through myristoylation of its N-terminus (Fig 1B). This configuration meant that on light-mediated dissociation of Zdk from the transmembrane component of the receptor, the signaling moiety would diffuse away laterally from the receptor–ligand complex and be rapidly dephosphorylated by CD45 phosphatase, which replicates the response to dissociation of the TCR from its pMHC ligand (Fig 1B). We preferred this intracellular control of receptor dissociation so that there was minimal disruption of cell conjugation due to loss of binding between receptor and ligand.

The ectodomain of the OptoCAR described above provides an orthogonal T-cell input that can be initiated on addition of a small molecule dimerizer drug (Fig 1C). The power of this approach is that it allows the process of cell–cell engagement and interface formation to be temporally separated from the initiation of receptor signaling. Dimerizer addition can then initiate signaling within the context of T cell–APC conjugates in a synchronized manner, which is essential to follow signaling dynamics at high temporal resolution. The integration of optical and chemical inputs within the OptoCAR presented a unique way to control both the intensity and duration of proximal signaling at the T-cell conjugate interface and follow the associated downstream response (Fig 1C). Importantly, optical control does not directly perturb any part of the binding

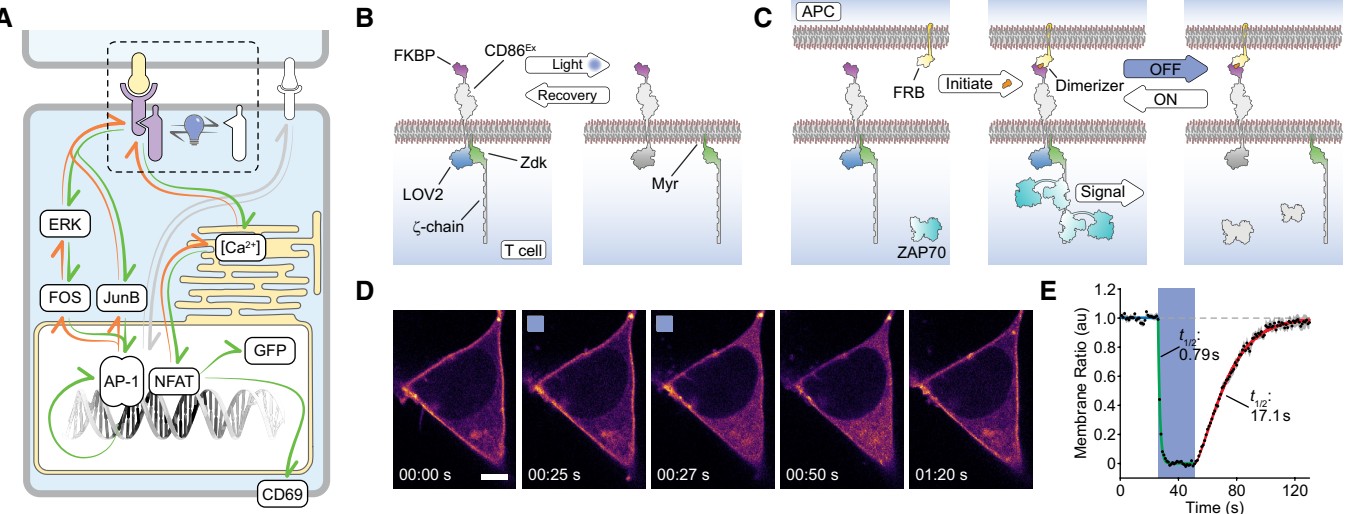

**Figure 1. Construction of an optically modulated chimeric antigen receptor (OptoCAR) in T cells.**

A  Schematic showing the OptoCAR replacing the antigen receptor (dotted region) as the primary driver of intracellular T-cell signaling. Illumination of the cells leads to the dissociation of the intracellular signaling domain of the OptoCAR complex and the loss of signal transduction. The network nodes investigated in this study are explicitly depicted within the boxes.

B  A more detailed schematic of the bipartite OptoCAR complex from (A) with component parts labeled. Light-mediated disruption causes dissociation of intracellular signaling tail (ζ-chain) from the extracellular binding domain.

C  OptoCAR-T cell engagement with an antigen-presenting cell (APC) does not initiate signaling in the absence of the dimerizer small molecule. OptoCAR signaling from within these cell conjugates is modulated by transitioning from the dark (signaling-competent) state to the "Off" state by illuminating cells with blue light. The intracellular domain will be rapidly dephosphorylated when it diffuses away from the bound OptoCAR.

D  Stills from Movie EV1 showing the membrane-bound form of the OptoCAR expressed in HEK293T cells being reversibly dissociated from the extracellular domain by single-cell illumination (blue box). The intracellular part of the OptoCAR used in this experiment was not myristoylated so it localized to the cytoplasm on dissociation, to aid visualization of OptoCAR dynamics. Scale bar: 5 μm.

E  Quantification of the light-mediated dissociation of OptoCAR in HEK293T cells by measuring the ratio of membrane-bound intracellular part of OptoCAR over time. Individual cells were illuminated after 25 s of imaging for 25 s to calculate dissociation rate (green) before returning to dark (signaling-competent) state to calculate re-association rate (red line). A unitary value was assigned to the mean membrane fluorescence prior to illumination and zero to the minimum value. Raw plots are presented in Appendix Fig S1A. Bounding area around data points shows mean ± SEM ($n = 12$ cells).

interface between the two cells and so engagement of any other costimulatory receptors, such as CD28, would remain unaffected. This allows us to focus on the integration capability of antigen receptor-initiated signaling in isolation.

We first wanted to directly test the response of the OptoCAR to light control. We expressed the receptor in HEK293T cells, which are non-immune cells and do not express proteins that normally interact with the TCR. In its normal bipartite architecture, the dissociation of the OptoCAR on light stimulation would simply alter the diffusion rate of the two components, which is difficult to quantify. We therefore modified the OptoCAR so that the intracellular fragment would dissociate and diffuse freely in the cytoplasm. This then allowed us to visualize the light-mediated dissociation of the OptoCAR by labeling the intracellular fragment with a red fluorescent protein (mScarlet) and measure its cellular localization by confocal microscopy.

Stimulation with blue light caused the rapid dissociation of the bipartite receptor (Fig 1D and Movie EV1) with an off-rate of 0.88/s, implying near-complete (95%) dissociation within ~3.5 s (Fig 1E and Appendix Fig S1A). Turning off the light caused the thermally induced reversion to the "dark" (signaling-competent) state of the LOV2 domain and hence reformation of the signal-competent OptoCAR (Fig 1E, Movie-EV1, and Appendix Fig S1A). This process

was slower with an on-rate of 0.04/s and near-complete complex association within ~75 s, which agrees very well with previously reported values for the LOVTRAP system (Wang *et al*, 2016). We also used a LOV2$^{C450G}$ variant (Wang *et al*, 2016) of the OptoCAR that is light-insensitive to rule out phototoxicity-induced dissociation. (Appendix Fig S1B).

## Activation-induced intracellular Ca$^{2+}$ flux is rapidly disrupted on signal cessation

Having shown that the OptoCAR itself was both light-responsive and could be reversibly modulated, we next wanted to test whether the OptoCAR functioned as anticipated in T cells to influence the proximal receptor signaling pathway. We used viral transduction to stably express the OptoCAR in the Jurkat T cell line (OptoCAR-T cells) and confirmed that the synthetic reporter was readily detectable at the cell surface (Fig EV1A–C).

One of the earliest direct readouts of T-cell activation is Ca$^{2+}$ fluxing, the rapidly increased concentration of Ca$^{2+}$ ions in the cytoplasm from ER-derived stores (Fig 1A). T cells expressing the OptoCAR were first loaded with Indo-1, a ratiometric Ca$^{2+}$ sensor, before being conjugated with a B-cell line expressing the counterpart ligand (Raji-FRB$^{Ex}$). Pertinently, the initial engagement between the

two cell types was driven solely by adhesion proteins in the absence of the dimerizer, which ensured that proximal signaling through the OptoCAR remained uninitiated, although other accessory receptors could still be engaged. Flow cytometry was then used to gate on only these cell conjugates prior to dimerizer addition (Fig EV1D and E) and follow changes in intracellular $Ca^{2+}$ concentration over time.

We first confirmed that no cellular response was detectable when a vehicle control (solvent alone) rather than the dimerizer was used to initiate OptoCAR-mediated signaling (Fig 2A). We then performed the assay in the "dark" state (without illumination) to assess whether OptoCAR engagement could initiate proximal T-cell signaling. The addition of dimerizer to the conjugated T cells did

indeed cause a robust increase in intracellular concentration of $Ca^{2+}$ ions, with the flux saturating within ~2 min (Fig 2B).

To provide optical control over receptor signaling while simultaneously measuring the calcium flux, we custom-built a heated LED illuminator that could be installed directly onto the flow cytometer (Fig EV1F). Repeating the assay above under constant blue light illumination completely inhibited $Ca^{2+}$ fluxing to an equivalent extent as the vehicle control (Fig 2C), showing that the OptoCAR was indeed light-responsive in T cells. To rule out that this observation was due to any phototoxic effects of light stimulation, we repeated the assay with the light-unresponsive OptoCAR[C450G] variant described above. In this case, the observed $Ca^{2+}$ flux was found to be maintained even under constant illumination (Fig EV1G).

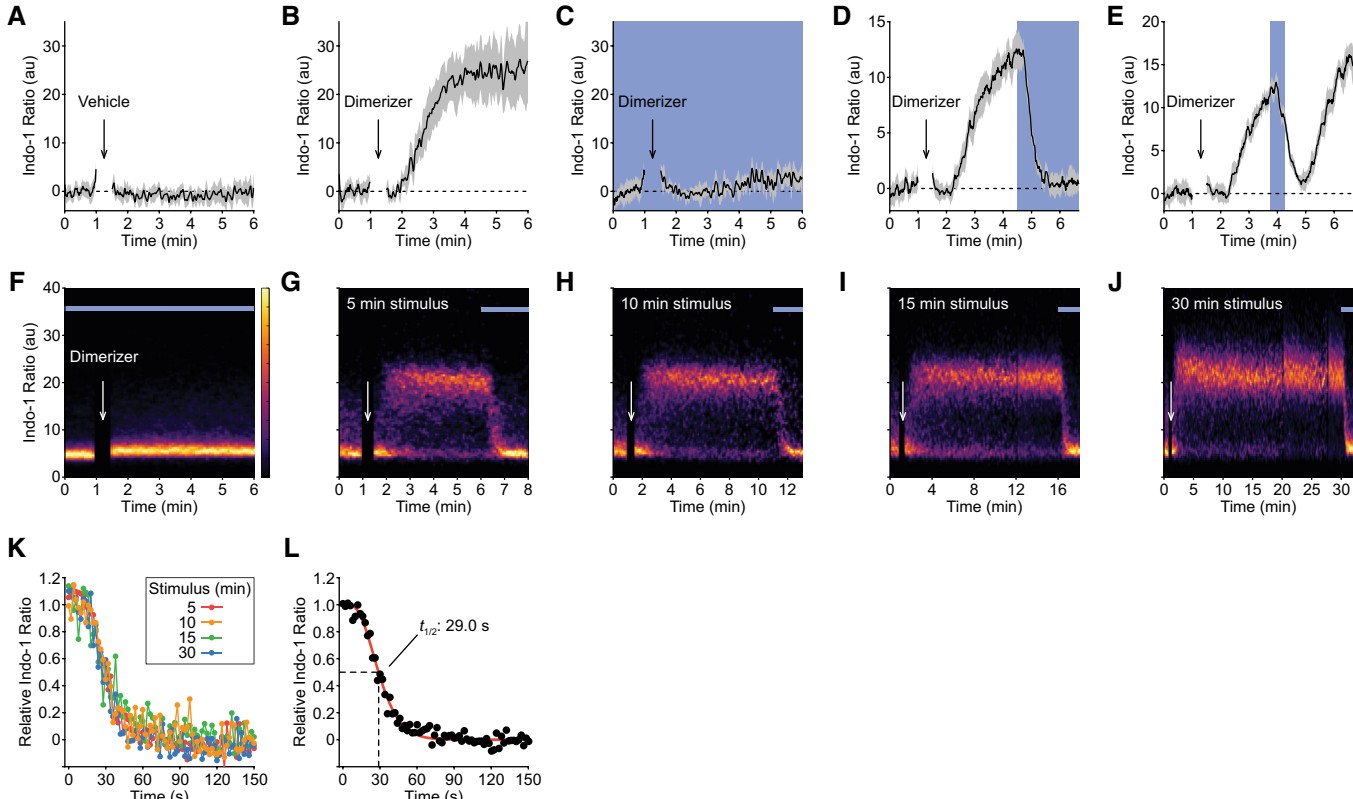

**Figure 2. Activation-induced intracellular $Ca^{2+}$ flux is rapidly disrupted on signal cessation.**

A   OptoCAR-T cells were loaded with a ratiometric $Ca^{2+}$ indicator (Indo-1) before conjugation with ligand-presenting cells. Conjugated cells were gated on by flow cytometry and the Indo-1 ratio, as a readout of intracellular $[Ca^{2+}]$ was measured over time. Vehicle addition after 60 s caused no detectable change in this ratio. Bounded line shows mean ± SEM ($n = 5$).

B   An equivalent experiment was set up as in (A), but now 2 μM dimerizer was added after 60 s. Conjugated cells were maintained in the dark to maintain OptoCAR signaling. Bounded line shows mean ± SEM ($n = 3$).

C   An equivalent experiment was set up as in (B), but the conjugated cells were continuously illuminated throughout dimerizer addition to disrupt OptoCAR signaling. Bounded line shows mean ± SEM ($n = 3$).

D   An equivalent experiment was set up as in (B), but now conjugated cells were illuminated 3.5 min after dimerizer addition. Bounded line shows mean ± SEM ($n = 4$).

E   An equivalent experiment was set up as in (B), but now cells were illuminated 165 s after dimerizer addition, for 30 s. Bounded line shows mean ± SEM ($n = 3$).

F–J   Representative density plots of $Ca^{2+}$ flux from conjugated OptoCAR-T cells over time at 37°C. Conjugated cells were stimulated with dimerizer addition for 0 (F), 5 (G), 10 (H), 15 (I), or 30 (J) minutes prior to disrupting signaling by illuminating cells.

K   The decrease in intracellular $[Ca^{2+}]$ on illuminating conjugated cells after different intervals during activation was plotted as the mean Indo-1 ratio ($n = 3$). The relative $[Ca^{2+}]$ was calculated by removing background Indo-1 ratio before scaling to maximal output. Inset legend delineates the datasets.

L   A single plot combining all datasets from (K) was fit by the survival function of a gamma distribution, with indicated half-life calculated from the time taken for the relative $[Ca^{2+}]$ to decrease to 50% of maximum.

Conversely, using a LOV2 mutant (I539E) that remains open even in the dark (Wang *et al*, 2016), we could not observe any fluxing (Fig EV1H), confirming that the OptoCAR was not simply creating a signaling-competent region within the plasma membrane for the endogenous TCR (or other receptors) at the cell surface to signal.

Having shown that the OptoCAR was responsive to light modulation, we wanted to use our new tool to investigate the kinetics and reversibility of how proximal signaling drives $Ca^{2+}$ fluxing. We repeated the experiment above with the light-responsive OptoCAR, but this time illuminated the cells once the intracellular concentration of $Ca^{2+}$ ions had begun to increase upon dimerizer addition. Illumination of the cells undergoing proximal signaling caused the rapid termination of the $Ca^{2+}$ flux, which reverted to baseline within approximately 30 s (Fig 2D). This light-mediated disruption of proximal signaling was readily reversible once illumination ceased and the OptoCAR reverted to the signaling-competent state (Fig 2E).

The extremely rapid decrease in intracellular $Ca^{2+}$ concentration on light-mediated cessation of receptor signaling was unexpected given that illumination of cells only disrupts the functional state of the receptor itself. We therefore questioned whether this result was because we had disrupted signaling so soon after its initiation and had not allowed proximal signaling to become self-sustaining, if it had this capacity. To address this point, we performed the equivalent assay but illuminated the conjugates to disrupt proximal signaling at different time points after its initiation. Constant illumination caused no detectable $Ca^{2+}$ flux in the raw cytometry plots as expected (Fig 2F), but light-mediated cessation of receptor activation caused a rapid decrease in $Ca^{2+}$ fluxing even 30 min after activation (Fig 2G–J). We quantified the dynamics of $Ca^{2+}$ flux disruption on illumination and found that all stimulation periods showed essentially equivalent kinetics of signal decay (Fig 2K). Thus, this part of the proximal signaling network remains rapidly reversible and is entirely contingent on continuous receptor signaling, with no discernible adaptation to more sustained proximal signaling. By fitting the combined datasets, we estimated that it takes ~8 s for the information of signal cessation to propagate to the $Ca^{2+}$ channels (1% decrease in output) and that the effective half-life of proximal signal within this part of the network to be 29 s (95% CI: 27.4–30.5 s) (Fig 2L).

Overall, the implication of this dataset is that the counteracting reactions along the signaling network from the receptor activation to the $Ca^{2+}$ stores in T cells are efficient at dissipating proximal signaling once the stimulus has been removed, presenting very limited signal persistence at this level.

### ERK activation remains sensitive to the state of proximal signaling

The previous results suggested that continuous signaling from the receptor is required to maintain the intracellular fluxing of $Ca^{2+}$ ions. We were therefore keen to understand whether this result held true for steps more distal from receptor activation in the T-cell signaling network. A well-studied part of this pathway is the activation of extracellular signal-regulated kinase (ERK) by dual phosphorylation at Thr[102] and Tyr[104] (Roskoski, 2012). ERK is a classical member of the mitogen-activated protein kinase (MAPK) family, and previous studies have suggested that ERK activation in T cells is digital (Altan-Bonnet & Germain, 2005; Das *et al*, 2009). We

therefore wanted to use our OptoCAR system to investigate how the activating phosphorylation of ERK is modulated by the disruption of upstream signaling. Conjugates between OptoCAR-T cells and Raji-FRB[Ex] cells were prepared in the absence of the dimerizer and placed in the blue light illuminator as used for the $Ca^{2+}$ flux experiments. The dimerizer was then introduced and sample aliquots were taken at defined time points by rapid fixation over a period of 30 min, with or without illumination (Fig 3A). Cells were then intracellularly stained for doubly phosphorylated ERK (ppERK), and the signal from conjugated cells was measured by flow cytometry (Fig 3A).

Carrying out the experiment with the OptoCAR in the dark (signal-competent) state led to a rapid initial phosphorylation of ERK, with ppERK detectable within 1 min before plateauing around 10 min (Fig 3B and E). The distributions of ppERK intensities over time show a clear all-or-nothing response to stimulation (Fig 3E), as previously reported (Altan-Bonnet & Germain, 2005). We were confident that this was the physiological response to receptor signaling because activation was synchronized by dimerizer addition, and we specifically gated on OptoCAR-T cells conjugated to ligand-presenting cells. We then repeated the experiment but now illuminated the conjugates to disrupt receptor signaling after 5 min and continued to measure the distribution of ppERK staining. Illumination caused a rapid decrease in ppERK staining, which was detectable within 30 s and continued to decrease to baseline levels (Fig 3C and F). We were able to extract the half-life of the ppERK modification on cessation of proximal signaling, which we measured to be 3.0 min (95% CI: 2.4–3.6 min). We confirmed that the light-induced decrease in ppERK was signaling-dependent using the OptoCAR[C450G] and OptoCAR[I539E] controls described above (Fig EV2).

Having shown that OptoCAR-mediated signaling was capable of activating ERK in a light-dependent manner, we wanted to investigate whether the observed inhibition of proximal signaling was reversible, as we had found for $Ca^{2+}$ fluxing (Fig 2E). After initiating T-cell signaling with the dimerizer, we pulsed the cell conjugates with light after 5 min for 3 min before returning them to the signaling-competent state. As anticipated, ERK phosphorylation resumed quickly after resumption of receptor signaling at a rate that was equivalent to that observed prior to illumination (Fig 3D and G). Importantly, ppERK levels increased from the previous point immediately after reverting to the dark state, demonstrating that signal persistence within the network can lead to a more rapid increase in ppERK. However, the measured half-life of 3 min for ppERK decay after cessation of proximal signaling implied that any biochemical "memory" at this part of the signaling network would completely dissipate by ~10 min.

### Activation of FOS transcription factor remains dependent on proximal signaling

The preceding dataset showed that the MAPK pathway remained rapidly reversible on termination of receptor signaling albeit with more persistence compared with the $Ca^{2+}$ flux dynamics. We reasoned that this reversibility may extend to transcriptional activation within the nucleus, a step that is significantly more distal to the cell surface activation of the OptoCAR (Fig 1A). Many transcription factors (TFs) require phosphorylation to become active and enhance gene expression. This is true for the AP-1 family of leucine zipper

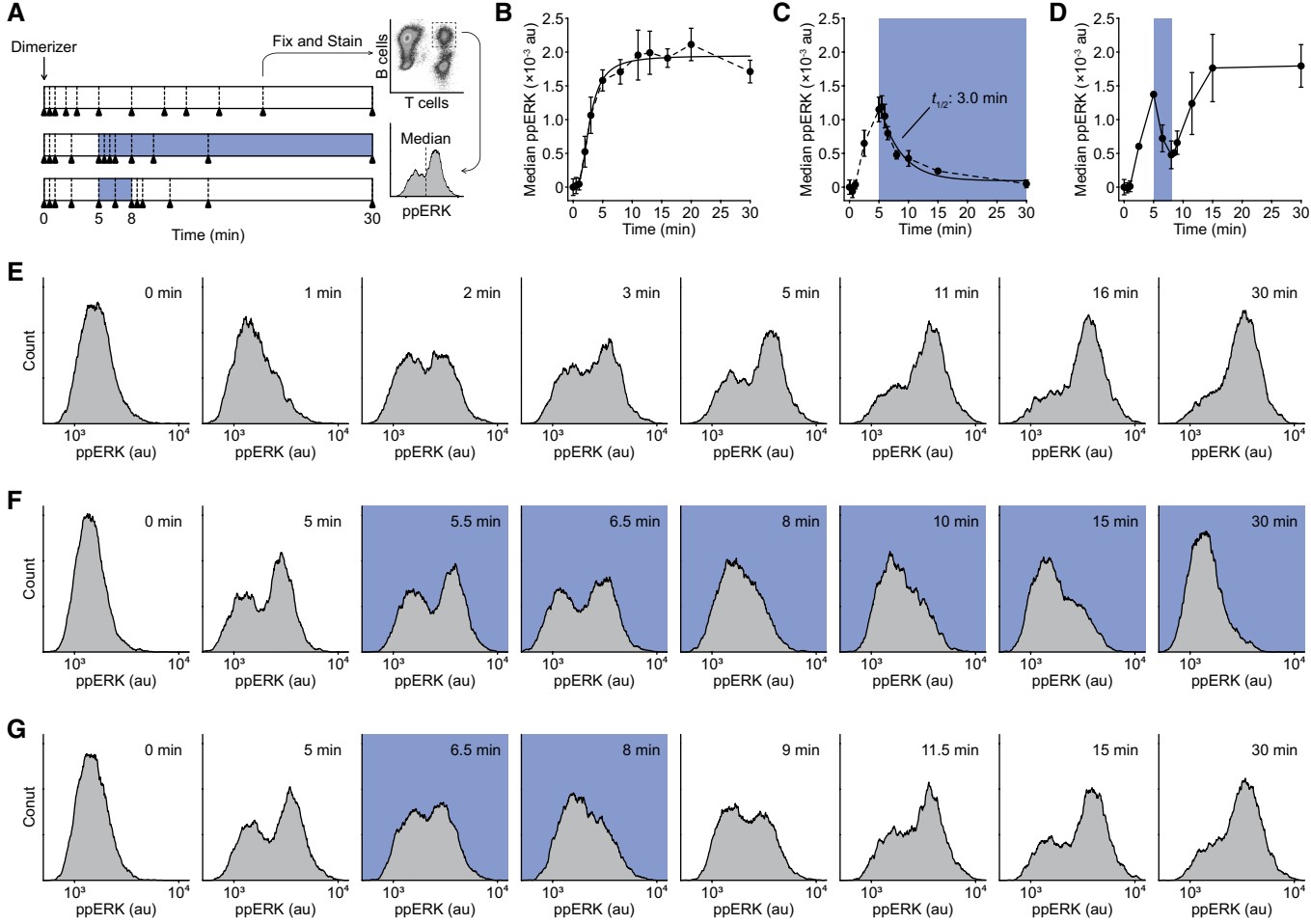

**Figure 3. ERK activation remains sensitive to the state of proximal signaling.**

A Schematic of the OptoCAR-T cell experiment to follow the dynamics of ERK phosphorylation. Dimerizer was added to cell conjugates at 0 min, with samples (denoted by triangles) taken at defined points. Where sample illumination was used, this is shown by blue region. Samples were rapidly fixed and stained for doubly phosphorylated ERK (ppERK) before measuring the distribution of ppERK levels in cell conjugates.

B OptoCAR-T cells were conjugated with ligand-presenting cells and activated as in the upper regime shown in (A). The median of ppERK intensity distribution was plotted with time, after subtraction of background intensity. Bars show mean ± SEM ($n = 3$).

C An equivalent experiment was performed as in (B) but following the middle regime shown in (A), with sample illumination after 5 min for the remainder of the dataset. Indicated half-life was calculated from the time taken for the median ppERK to decrease to 50% of maximum. Bars show mean ± SEM ($n = 4$).

D An equivalent experiment was performed as in (B) but following the lower regime shown in (A), with sample illumination after 5 min for 3 min. Bars show mean ± SEM ($n = 3$).

E–G A subset of ppERK distributions from individual datasets were used to create plots in (B–D), respectively. Blue boxes denote samples that were illuminated.

TFs, typified by the FOS/JUN heterodimer (Karin *et al*, 1997). In T cells, FOS protein is rapidly expressed on activation as an immediate-early gene (Ullman *et al*, 1990; Clark *et al*, 2011) and phosphorylation increases its TF activity to function, in conjunction with JUN, to upregulate the expression of many genes (Karin *et al*, 1997).

We investigated the dependence of FOS phosphorylation on upstream signaling by conjugating the OptoCAR-T cells with ligand-presenting cells as normal in the presence of the dimerizer. We first measured the kinetics of *FOS* expression and its phosphorylation in the dark state to be sure that OptoCAR signaling could drive the expression and activation of nuclear TFs. OptoCAR-T cells were activated by dimerizer addition to initiate signaling, and samples were then taken at defined time periods, rapidly lysed, and subjected to

fluorescent Western blot analysis. Both the phosphorylation of FOS at Ser[32] (Figs 4A and EV3A) and the total levels of FOS protein (Figs 4B and EV3B) were quantified, with the Ser[32] modification known to correlate with increased FOS TF nuclear localization and stability (Sasaki *et al*, 2006). We found that FOS protein was undetectable prior to activation in T cells but readily induced within 30 min (Fig 4B and C), and phosphorylation at Ser[32] was detectable within 60 min (Fig 4A and D). Because neither the OptoCAR-T cells nor the Raji-FRB[Ex] cells expressed FOS prior to activation, we could be confident that the detected protein arose solely from T-cell activation. Total FOS protein peaked 2 h after activation before decreasing (Fig 4C). However, quantifying the abundance of the Ser[32] phosphorylated state as a fraction of total FOS showed that the

nuclear-localized version continued to accumulate with increased signaling duration (Fig 4D). This effect was not due to the fraction of cells that were conjugated, which remained stable 60 min after dimerizer addition (Fig EV3C). Performing an equivalent experiment using the OptoCAR$^{I539E}$ variant that cannot form the signaling-competent receptor showed no FOS expression as expected (Fig EV3D).

Next, we investigated how light-mediated disruption of the OptoCAR in activated T cells controlled the fraction of active c-FOS TF. The primed active form of FOS runs as a higher molecular weight band by electrophoresis, compared with the unphosphorylated

form (Murphy *et al*, 2002), and we used this feature to measure the fraction of functional FOS in our Western datasets (Fig 4A). After 3 h of continuous stimulation in the dark, OptoCAR-T cell conjugates were illuminated using the optoPlate (described below) and samples were then taken at defined time points over the next 15 min. The disruption of OptoCAR-mediated signaling caused a detectable loss of the higher molecular weight phospho-FOS band (Fig 4A and E). Plotting the relative abundance of the activated form (Fig 4F) with time showed that the higher molecular weight FOS TF was lost to a baseline value after 10 min (Fig 4G). Fitting these individual datasets after normalization allowed us to quantify the

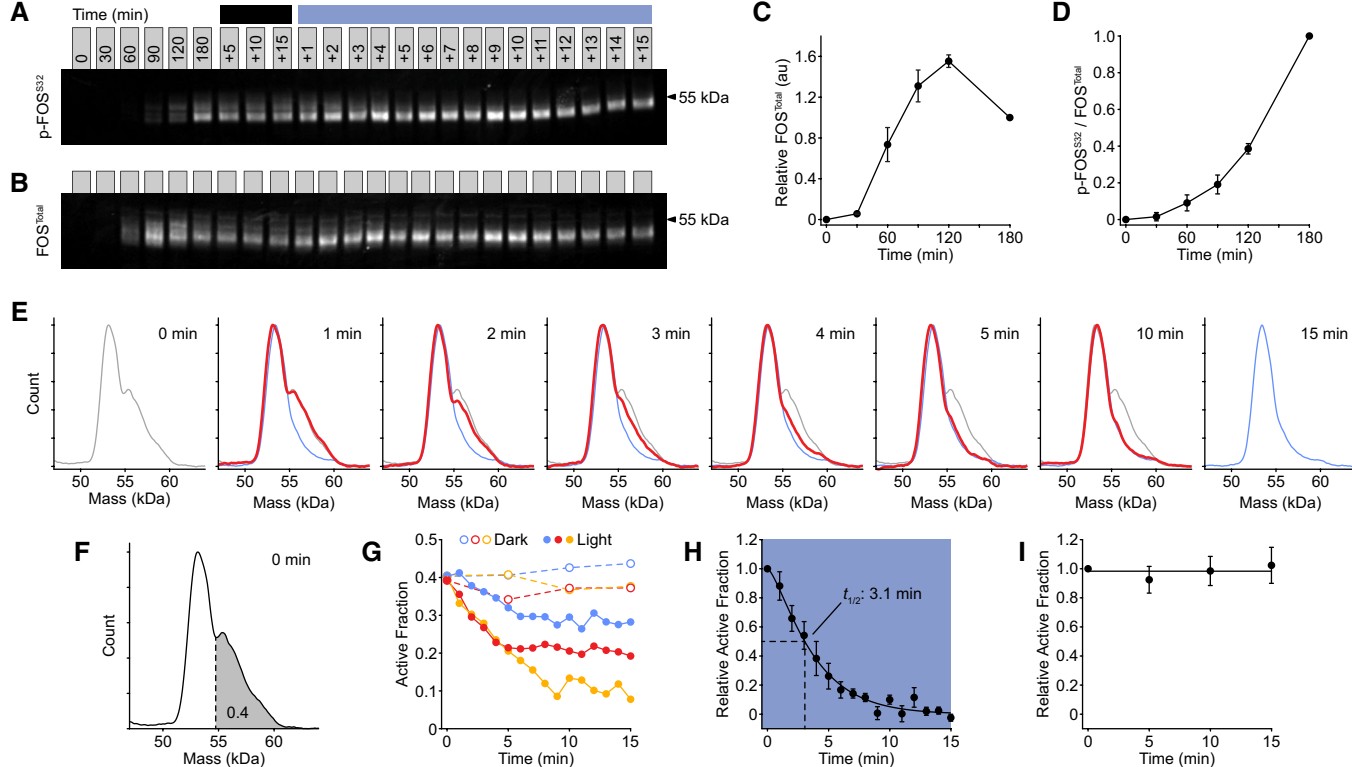

**Figure 4. Activation of FOS transcription factor remains dependent on proximal signaling.**

A  Representative fluorescent Western blot showing the dynamics of FOS phosphorylation (p-FOS$^{S32}$) on light-modulated control. Conjugated OptoCAR-T cells were stimulated for a defined period after dimerizer addition (denoted above blots) before either being left in the dark (dark line) or illuminated (blue line). Full blot image with total protein normalization control shown in Fig EV3A.

B  Representative fluorescent Western blot showing the dynamics of FOS expression (FOS$^{Total}$) on light-modulated control, from the same dataset as in (A). Full blot image with total protein normalization control shown in Fig EV3B.

C  Quantification of FOS expression detected by fluorescent Western blots at different time points prior to illumination. Datasets were corrected for background fluorescence and scaled to FOS level at 3 h. Bars show mean $\pm$ SEM ($n = 3$).

D  Quantification of the magnitude of phosphorylated FOS (p-FOS$^{S32}$) as a fraction of total FOS, detected by fluorescent Western blots at different time points prior to illumination. Datasets were scaled relative to phosphorylated FOS level at 3 h. Bars show mean $\pm$ SEM ($n = 3$).

E  Distributions of phosphorylated FOS molecular mass at different time points after illuminating OptoCAR-T cell conjugates to disrupt signaling, quantified from blot in (A). The molecular mass distributions on initial illumination after 3 h of activation (0 min) and end of measurement (15 min) are shown as gray and blue lines, respectively. Distributions in other plots are shown as red lines that vary between these extremes.

F  A schematic to show how the active (higher molecular mass) fraction was estimated from the distributions in (E), with a cutoff of 0.4 defined by the 0 min distribution and used to analyze the datasets.

G  The data from three independent experiments (blue, red, and yellow) are shown either when FOS active fraction was measured in the dark (dashed lines) or under illumination (solid lines).

H  The light state datasets shown in (G) were combined and rescaled between the active fraction at 0 and 15 min. Indicated half-life was calculated from the time taken for the activated fraction to decrease to 50% of maximum. Bars show mean $\pm$ SEM ($n = 3$).

I  The dark state datasets shown in (G) were similarly combined and rescaled, as in (H). Bars show mean $\pm$ SEM ($n = 3$).

half-life decay of FOS activity on cessation of proximal signaling, which we found to be 3.1 min (95% CI: 0.7–5.5 min) (Fig 4H). The paired samples that remained in the signaling state showed no change in the active FOS fraction (Fig 4I). To ensure that this rapid decrease in TF activity on illumination was due to signal dissipation, we measured the active FOS fraction over the same 15-min period using the OptoCAR$^{C450G}$-T cells and saw no significant difference between the dark and light states (Fig EV3E).

Overall, we have shown that the activation state of an important TF required for T-cell activation remains intimately constrained by the signaling potential of the cell surface receptor, and confirmed the results above that the opposing steps in the signaling pathway that return the system to equilibrium are very efficient.

## Gene transcription is rapidly abolished after disruption of receptor signaling

The weak persistence of active FOS on disruption of receptor signaling agreed well with the equivalent dynamics of ppERK dephosphorylation, suggesting that much of the intracellular signaling network remains intimately connected to the functional state of the upstream receptor. The functional output of TFs is the production of mRNA transcripts encoding for a set of genes. Thus, we wanted to ascertain whether the rapid disruption of FOS and ERK activity was also observed at the mRNA level or whether transcript production showed significant persistence even on cessation of signaling. To explore this question, we measured the relative mRNA concentration of several genes upregulated during OptoCAR-T cell activation using reverse transcription quantitative PCR (RT-qPCR). We chose to quantify the mRNA from genes with a range of distinct TF requirements in Jurkat T cells: CD69 is predominantly activated by AP-1 (Castellanos *et al*, 1997), efficient expression of the cytokine interleukin-2 (IL2) is dependent on CD28-mediated NFAT and AP-1 TFs (Shapiro *et al*, 1997; Spolski *et al*, 2018), CXCL8/IL8 transcription requires NF-κB activity (Kunsch & Rosen, 1993; Hoffmann *et al*, 2002), and FOS transcription is dependent on Elk-1, which is downstream of ERK activity (Cavigelli *et al*, 1995).

We initiated signaling in conjugated OptoCAR-T cells with the dimerizer drug as normal and took samples over the next 3 h. We then either illuminated the OptoCAR-T cells and collected samples for a further 60 min or left them under dark conditions to maintain signaling. The genes investigated presented a range of expression profiles, with *FOS* showing rapid expression as required for an immediate-early gene, and *IL-2* a delayed response that may be due to combining signals from the OptoCAR and CD28 costimulatory receptor (Fig 5A and Appendix Fig S2). However, all mRNA transcripts showed a pronounced decrease in illumination that approached unstimulated levels over 60 min (Fig 5B and Appendix Fig S2). While there appeared to be continued gene transcription for CD69 and IL-2 after 3 h of receptor signaling (Fig 5A), *FOS* and *CXCL8* expression were decreasing prior to illumination, due to the normal regulation of their expression profiles. Pertinently, mRNA production for these genes was still dependent on the signaling state of the OptoCAR. Correcting for the underlying trajectory of mRNA expression by taking the ratio between the paired illuminated and dark samples, we found that mRNA levels started to decrease for all genes within 5 min of illumination (Fig 5C). This result implied that nascent mRNA production is halted at an equivalent timescale to inhibition of TF activity, with the

observed decay presumably due to mRNA degradation pathways, assumed to be independent of the active state of the promoter. Thus, these decay rates with approximate half-lives of 15–30 min correspond to the limiting bound on how long intermediary signals are sustained within the network.

## Short-term signal persistence can be directly observed in gene expression output

The previous results pointed to the rapid disruption of intracellular signaling when OptoCAR activity is removed, which implies that any persistent state within the network could only have an effect on the minute timescale. Nonetheless, it is feasible that this short-term biochemical memory could play a role in integrating signaling pulses that are temporally discrete into an output response greater than the sum of the individual stimuli (Fig 6A).

To first confirm that OptoCAR-mediated signaling did indeed result in detectable protein outputs, we measured the *de novo* gene expression from two readouts of T-cell activation (Fig 1A). The first output was NFATc1-mediated GFP expression, where NFATc1 (NFAT) is a TF that normally combines with AP-1 TFs to drive expression of key proteins critical for T-cell activation (Hogan, 2017), including IL-2. The second output is the increased expression of CD69 at the cell surface (Cibrian & Sanchez-Madrid, 2017). OptoCAR-T cells were conjugated as in the previous experiments while either providing continuous illumination or keeping them in the dark (signal-competent) state, and NFAT-mediated GFP expression and CD69 upregulation were then measured. As expected, OptoCAR-T cells activated in the dark for 24 h showed potent upregulation of both functional outputs, but responses were essentially undetectable under constant illumination during activation (Fig EV4A).

To perform these experiments, we employed a modified version of an "optoPlate" that is capable of independently illuminating the wells of a 96-well plate with control over both the duration and intensity of light pulses (Bugaj & Lim, 2019). We used an intensity of light that ensured signal quiescence (Fig EV4B) while minimizing phototoxicity, measured using the light-unresponsive OptoCAR$^{C450G}$ receptor (Fig EV4C and D). We also utilized the OptoCAR$^{V416L}$ receptor variant for these longer-term experiments to further minimize any deleterious effects of continuous illumination (Wang *et al*, 2016). We used live-cell confocal microscopy to directly measure the kinetics of NFAT-mediated GFP expression within cell conjugates. It took approximately 90 min for GFP fluorescence to become significantly ($P < 0.05$) detectable above background (Fig EV4E and Movie EV2), which would be expected from the combined requirements for maturation of the fluorophore and *de novo* gene expression.

Having shown that OptoCAR activation could drive efficient gene expression, we next explored the relationship between the duration of a single pulse of receptor signaling and the magnitude of the downstream outputs. Conjugated OptoCAR-T cells were maintained in the dark to drive signaling for a given duration before disrupting OptoCAR signaling by illumination for the remainder of 24 h, when all samples were collected simultaneously. This experiment is therefore not a time course but a measure of the impact of signal duration on overall output. We found that increased duration of signaling resulted in greater output for both NFAT-mediated GFP expression (Figs 6B and EV4F) and CD69 upregulation (Figs 6C and EV4F) over more than 12 h of stimulation, demonstrating that our system was

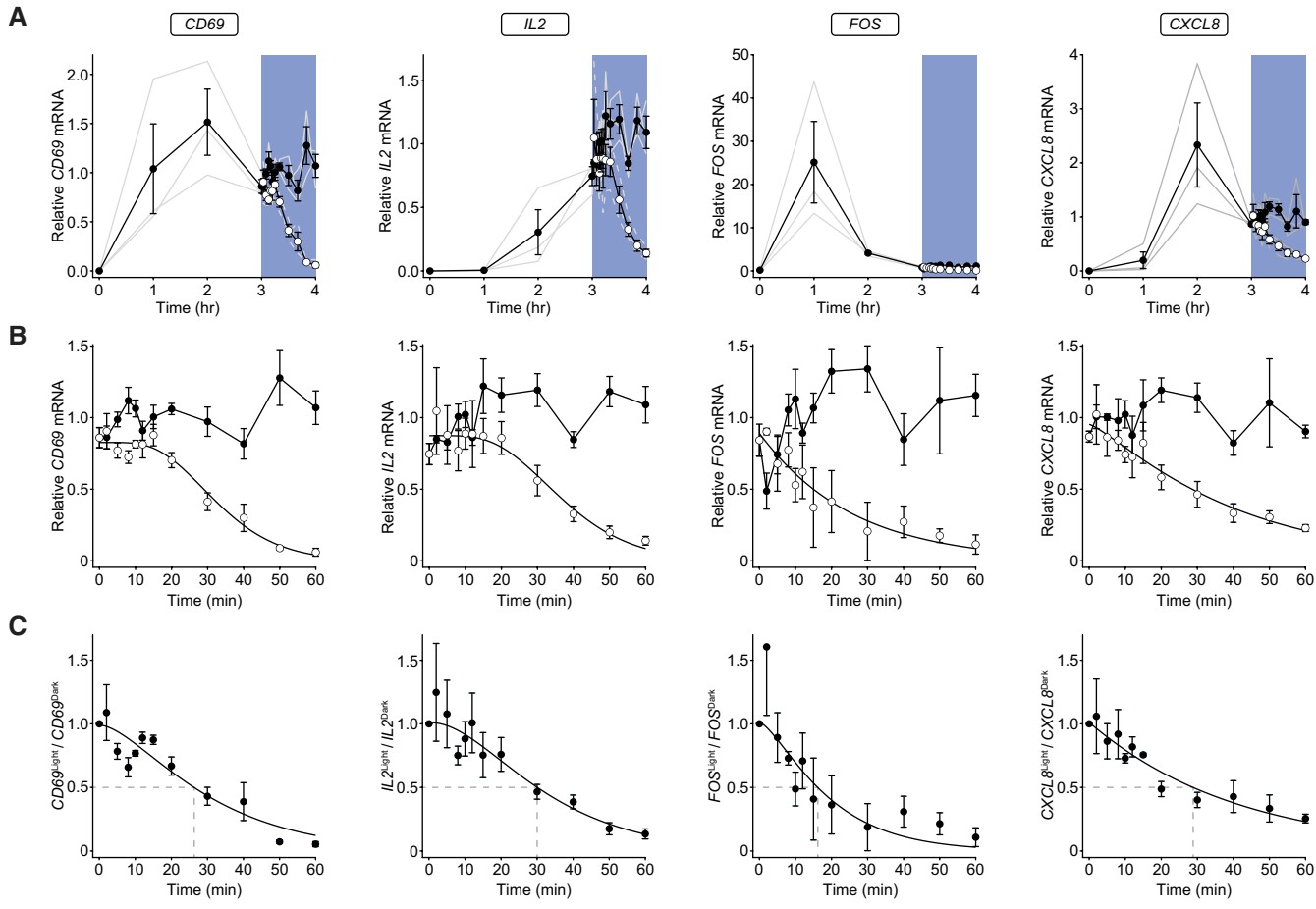

**Figure 5. Gene transcription is rapidly disrupted on cessation of OptoCAR signaling.**

A  RT-qPCR analysis of relative mRNA levels for four representative genes following OptoCAR-mediated T-cell activation. OptoCAR-T cell conjugates were activated in the dark (signaling-competent) state for a period of 3 h before light-mediated disruption of signaling (open circles) or maintained for a further 60 min in the dark (filled circles). The relative mRNA values were calculated by subtracting baseline value before scaling to mean output between 3 and 4 h. Individual datasets are presented as gray lines, with bars showing mean ± SEM ($n = 3$). Raw cRQ plots for all datasets are provided in Appendix Fig S2, and RQ values can be found in Dataset EV1.

B  The datasets from (A) are expanded to show the effect of illumination on the relative mRNA level of designated genes. As above, datasets are normalized to the mean mRNA level for the dark state values over the 60-min period. Bars show mean ± SEM ($n = 3$).

C  The ratio of relative mRNA in the light compared with dark state was calculated for each time point. An approximate half-life for mRNA decay is shown with dotted lines. Bars show mean ± SEM ($n = 3$).

not saturating within this period. The duration of the minimal pulse that caused a significant response above background ($P < 0.05$) was at least 30 min for both outputs, with CD69 upregulation showing the clearest response under brief stimulation.

For this, and all subsequent experiments, we used a fixed endpoint of 24 h to measure OptoCAR-T cell output. This ensured that signaling pulses were provided with sufficient time for potential output to be generated, as well as controlling for the total light exposure of different stimulation conditions. To confirm that this approach would not cause significant loss of output during extended illumination periods, we measured the decay rate of both outputs in activated cells. OptoCAR-T cells were stimulated for 16 h before output generation was blocked, either by illumination or by cycloheximide treatment to inhibit protein synthesis. We found half-lives of 24 and 14 h for GFP and CD69 degradation, respectively (Fig EV4G), demonstrating their longevity within cells. We also

followed output expression after signal cessation and similarly saw plateauing of GFP and CD69 expression (Fig EV4H).

As explained above, if signals can persist within the intracellular network after receptor dissociation, then there is the potential for integration of signal pulses into a more robust downstream response (Fig 6A). We therefore compared the output from a single 6-h period of sustained signaling with a sequence of discrete pulses, ranging from 15 to 120 min in duration but always totaling 6 h, separated by a varying light interval (Fig 6D). This is akin to pulse-frequency modulation, where we use a duty cycle ranging from 100 to 25%. We anticipated the measured output after 24 h to range between a maximum of continuous activation and the sum of independent pulses (Fig 6A). By encoding these pulsatile stimulation profiles onto conjugated OptoCAR-T cells, we could readily observe a light interval-dependent decrease in both NFAT-mediated GFP expression (Fig 6E) and CD69 upregulation (Fig 6F). Pertinently,

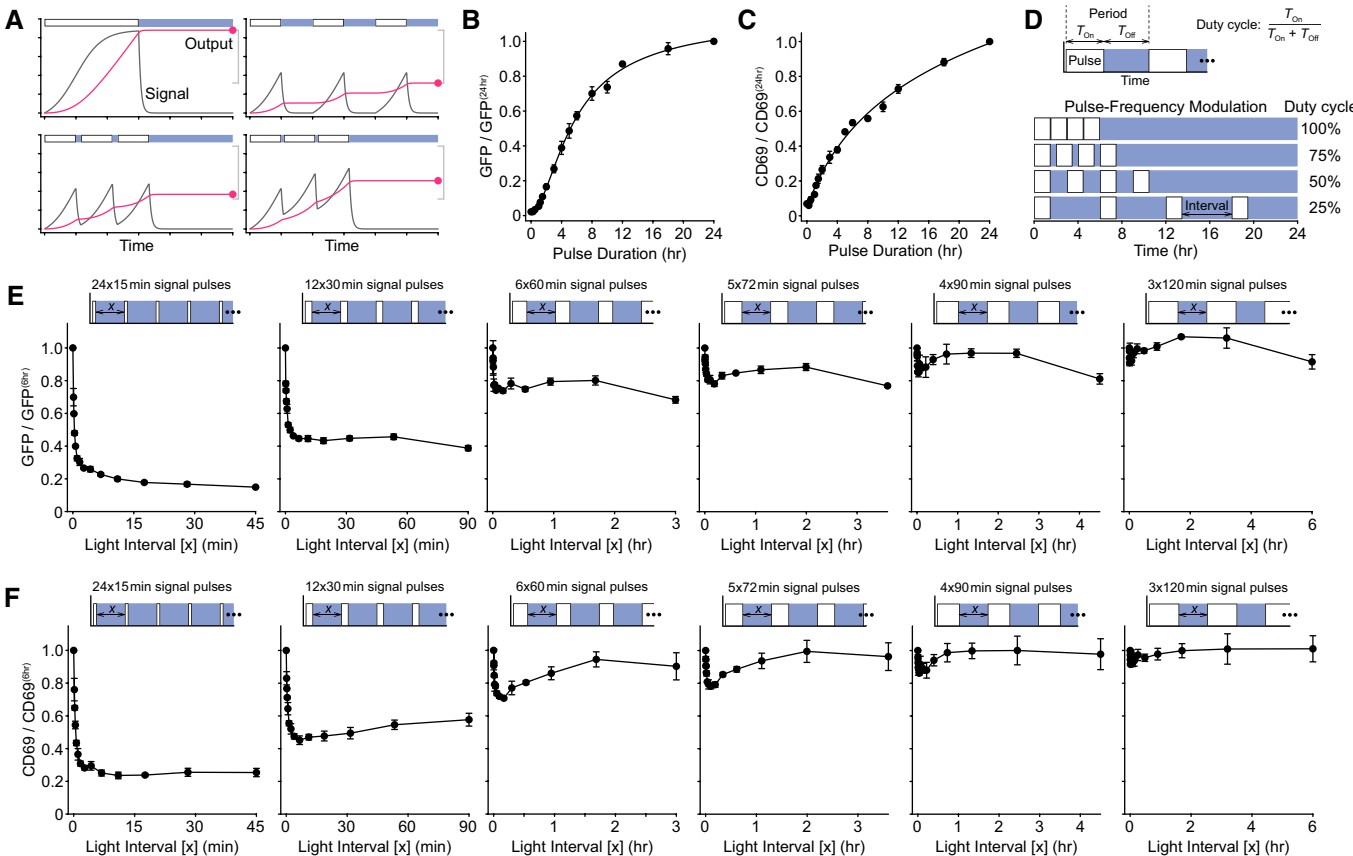

**Figure 6. Short-term signal persistence can be directly observed in gene expression output.**

A   Schematic showing the relationship between a signal (gray) and the corresponding output (red). A continuous signal pulse leads to maximal output (top left), whereas well-separated pulses independently increase output (top right). When pulses are separated by shorter periods when signals persist, the combined output is increased between these two bounds (bottom row).

B   Single pulses of varying duration were applied to OptoCAR-T cells, and the geometric mean of GFP expression was measured for all pulse lengths 24 h after initiation of signaling, with illumination started once the signal pulse had ended. All values have been corrected for background GFP fluorescence from non-activated OptoCAR-T cells and shown relative to the output at 24 h. A representative dataset is provided in Fig EV4F. Bars show mean ± SEM ($n = 3$)

C   Equivalent datasets as in (B) for the geometric mean of CD69 upregulation. A representative dataset is provided in Fig EV4F. Bars show mean ±SEM ($n = 3$)

D   Schematic illustrating the pulse-frequency modulation of signal pulses in OptoCAR-T cells. For a given pulse duration, the length of the light interval between pulses was varied between duty cycles of 100% (continuous) and 25%. The output after 24 h was measured for all samples.

E   Series of experiments showing how NFAT-mediated GFP expression is modulated by pulsatile trains of signaling. A combined signaling period of 6 h was broken into signal pulses ranging from 15 min to 2 h and the light-induced "off" interval between these pulses varied, denoted as x in the schematic above each plot. Geometric mean of GFP intensity is shown plotted as a function of light-induced "off" interval, relative to a single, continuous 6-h output. Bars show mean ± SEM ($n = 3$) of biological replicates.

F   Series of experiments showing how CD69 upregulation is modulated by pulsatile trains of signaling, from same dataset as in (E). Bars show mean ± SEM ($n = 3$).

this dependence decayed rapidly within 10 min for all stimulus periods (Fig EV4I), in excellent agreement with the kinetics of loss of active ERK (Fig 3C) and FOS (Fig 4H) TFs on signal cessation. We also repeated the assay over a 12-h window (rather than 24-h) and observed very similar results (Fig EV4J and K). We attribute the slight curving of the plateaus at longer light intervals to the slow degradation of pulse outputs (Fig EV4G), which would be less pronounced when pulses are closer to the end of the experiment. This dataset is strong evidence that persistence of residual signaling within the intracellular signaling network can be directly observed at the level of transcriptional output, but that it rapidly dissipates. However, while the overall output measured with pulsatile stimulation decreased rapidly when the inter-pulse interval increased over 10 min, it

plateaued to a level significantly higher than that expected for the sum of individual equivalent length single pulses. This was especially evident for the 15-min pulse datasets, where a single pulse gave an undetectable response, but repeated pulses became detectable. This implied that an additional intermediate within the signaling network could persist for longer between pulses, which could be explained by the slower decay of mRNA levels (Fig 5C).

## Graded signaling input to OptoCAR-T cells drives both analogue and digital outputs

From the experiments where a single pulse of varying duration was applied to conjugated OptoCAR-T cells, the mean output increased

with longer pulses in a continuous fashion (Fig 6B and C). However, when the single-cell distributions of these outputs were plotted, it suggested that longer signaling durations increased the fraction of cells that responded rather than a uniform increase in the outputs (Fig EV5A and B). We explored the nature of this response in more detail by titrating the input "strength" emanating from the OptoCAR and measuring the output distribution. This was achieved by continuously illuminating cells in each well of the optoPlate over 24 h but modulating the light intensity to quantify the cell's output response to this linear input (Fig EV5C). For both NFAT-mediated GFP expression (Fig EV5D and Movie EV3) and CD69 upregulation (Fig EV5E and Movie EV3), there was an initial regime under high illumination (low signaling) where output was undetectable. However, at a certain threshold of light intensity, OptoCAR-T cell activation became almost digital in nature, with the activated fraction of cells being dependent on signaling intensity. After this bifurcation, further increases in signal strength led to a more analogue increase in output. A higher-resolution dataset of this bistability in NFAT-mediated GFP expression with graded input provided a much clearer representation of the digital nature of this transition (Fig EV5E and Movie EV4) and suggested that some minimal level of signaling output must be reached before a cell becomes activated.

### Pulsatile signaling enhances the output of a clinically relevant anti-CD19 CAR

A sustained and potent stimulus can drive negative feedback loops within signaling networks (Amit *et al*, 2007), and there is evidence that a pulsed signaling input might alleviate this signaling-induced negative functional state (Wilson *et al*, 2017). Excessive or prolonged receptor signaling can drive primary T cells into a dysfunctional or "exhausted" state (Blank *et al*, 2019) and is commonly found with chimeric antigen receptors (CARs) currently used clinically (CAR-T therapy) to treat blood cancer patients (Wu *et al*, 2020). We therefore wanted to use our optically controlled receptor strategy to test whether pulsed CAR input might drive more effective T-cell activation. To this end, we engineered optical control into a clinically relevant anti-CD19 CAR (Porter *et al*, 2015), forming OptoCAR$^{CD19}$ (Fig 7A) and transduced primary CD4$^+$ T cells with this construct. We first confirmed that this modification to the CAR maintained its function by conjugating these OptoCAR$^{CD19}$ expressing T cells with a CD19$^+$ B-cell line and measuring expression of CD137 (4-1BB), which is a robust marker of T-cell activation (Wolfl *et al*, 2007) (Fig 7B). As expected, we found that continuous blue light illumination completely abrogated CD137 expression (Fig 7B).

We next varied the duration of a single pulse through the Opto-CAR$^{CD19}$ and measured CD137 expression after 24 h. In contrast to the equivalent experiment with the chemically inducible OptoCAR (Fig 6B and C), while there was an initial correlation between signal duration and cellular output, this plateaued after ~2 h with only a slow increase in output for pulses up to ~12 h (Fig 7C). Using RT-qPCR, we found no evidence for a pre-existing pool of *CD137* mRNA that might account for this initial burst of CD137 expression (Fig 7D). We hypothesized then that this outcome was due to signaling-induced feedback inhibition of CD137 expression and that by using short pulsatile bursts of signaling we might be able to alleviate this inhibition. As before, we split a single 6-h stimulus into pulses ranging from 15 min to 2 h and measured CD137 expression

after 24 h (Fig 7E). The response to pulsed signaling was very pronounced, with twenty-four 15-min pulses leading to an approximately threefold increase in output when compared to the continuous 6-h pulse and approached CD137 expression found after a continuous 24-h period (Fig 7F).

## Discussion

In this study, we have described an engineered antigen receptor that provides optical control over intracellular signaling in T cells while conjugated with antigen-presenting cells. We have used this new tool to directly interrogate the intracellular network's capacity for antigen receptor signal integration by quantifying the persistence of stimuli at various points within this network. The main finding from this work is that at all parts of the intracellular signaling network tested, disruption of antigen receptor signaling caused essentially all residual information within the network to dissipate within 10–15 min. This reinforces the view that proximal signaling from the TCR, and likely many other immune cell receptors, must be sustained for many hours to drive efficient activation, as previously suggested (Huppa *et al*, 2003), and likely requires the presence of costimulatory signals (Trendel *et al*, 2019). Nevertheless, we could directly observe the persistence of this biochemical signal within the network at the gene expression level using pulsatile trains of signaling. We also demonstrated that sustained proximal signaling is required to maintain transcription factors in an active state, and hence continued gene expression, with the decay of mRNA being the longest-lived signaling intermediate having approximate half-lives between 15 and 30 min for the transcripts measured here. We then demonstrated that this limited signal persistence could be exploited to increase the effective output from T cells when stimulated by a clinically relevant anti-CD19 CAR through pulsatile signaling.

Our results imply that T cells can only directly integrate TCR signals over multiple APC interactions within a short temporal window. As noted above, the interval between sequential T cell/APC interactions may be 1.5–2 h (Gunzer *et al*, 2000), which is incompatible even on the timescales we have measured for mRNA persistence. However, there is good evidence that T cells do accumulate the output from gene expression over multiple interactions (Faroudi *et al*, 2003; Munitic *et al*, 2005; Clark *et al*, 2011), which suggests that a threshold of protein expression must exist beyond which the specification of T-cell function occurs. By titrating the magnitude of the signal intensity emanating from the OptoCAR using graded illumination, we were able to demonstrate that a minimal level of signaling is required to drive downstream output in a digital manner, in agreement with this. The study from Bousso and colleagues (Clark *et al*, 2011) suggested that the accumulation of activated FOS was a potential mechanism for integrating multiple T-cell signaling events, where they found phospho-FOS$^{T325}$ increasing even when receptor signaling is disrupted. However, this result does not fit with the rapid loss of this modification when ERK activity is inhibited (Murphy *et al*, 2002), in agreement with our finding that FOS activity is rapidly lost on cessation of signaling (Fig 4F). This discrepancy may lie in how activated FOS was detected or the use of antibody-coated beads to activate the T cells. A limitation of our OptoCAR approach, and for CARs in general, is that there may be

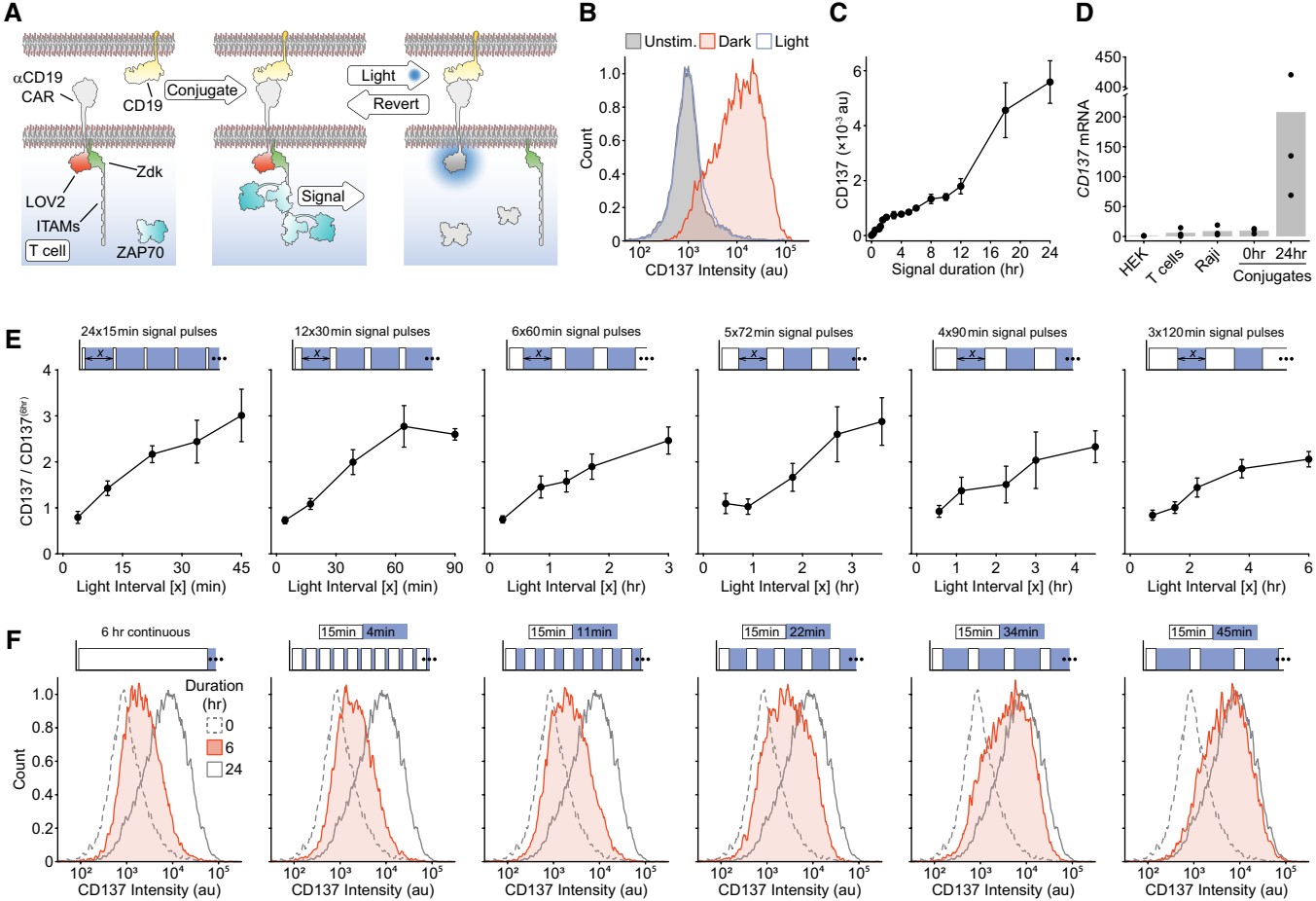

**Figure 7. Anti-CD19 CAR-T cell activation can be enhanced by pulsatile signaling.**

A   Schematic showing the anti-CD19 OptoCAR engineered to be light-responsive. Binding of the OptoCAR[CD19] expressed in T cells to CD19-expressing B cells drives signaling through an equivalent intracellular sequence used in the OptoCAR, where illumination with blue light causes the reversible disruption of receptor signaling.

B   Stimulating primary CD4[+] T cells expressing OptoCAR[CD19] over 24 h in the dark drives expression of CD137 (4-1BB), a robust activation marker. Continuous illumination completely abolishes this to CD137 levels equivalent to unstimulated T cells.

C   Plot of geometric mean of CD137 expression when the length of an individual signaling pulse is varied. Single pulse ranged from 0 to 24 h, with sample illumination initiated at the end of the pulse and CD137 expression measured for all samples 24 h after initiation of signaling. Bars show mean $\pm$ SEM of biological replicates ($n = 4$).

D   The cRQ values from RT-qPCR of indicated cells. Total RNA was extracted and CD137 mRNA levels measured relative to that from HEK293T cells. Conjugated OptoCAR[CD19]-T cells were also assayed either before or after 24 h stimulation in the dark state. Individual values from three biological replicates are shown, with mean depicted by bar plot.

E   Series of experiments showing how CD137 expression is modulated by pulsatile trains of signaling in primary CD4[+] T cells activated through OptoCAR[CD19]. A combined signaling period of 6 h was broken into pulses ranging from 15 min to 2 h and the refractory period between these pulses varied (x-axis), shown in the schematic above each plot. Geometric mean of CD137 intensity is shown plotted as a function of inter-pulse interval, scaled to continuous 6-h output. Bars show mean $\pm$ SEM ($n = 5$).

F   Representative dataset from (E), demonstrating how pulsatile stimulation of OptoCAR[CD19]-T cells drives more efficient activation. Dotted and gray histograms show CD137 expression in resting cells and 24-h activation, respectively. The filled histogram shows either a continuous 6-h stimulation (left) or cumulative pulsed signals over 24 h driving substantially more potent activation.

some other protein interactions governed by the accessory CD3 chains not present in these chimeric receptors that might help to sustain signaling even when they dissociate from ligand.

There was a clear correlation between the distance of the network intermediate from the activating receptor and the decreased rate at which signals were dissipated on disruption of signaling, in part explainable by the transition from Tyr- to Ser/Thr-based kinase reactions. This suggests that the duration of signaling might be

encoded by differential signal persistence within the network, with protein expression being the only state that can persist on the hour timescale. Given that many signaling networks are built from sequential steps, a better "memory" of receptor activation would become imprinted when more distal steps are stimulated. This would also have the effect of filtering out more spurious signaling pulses, which would be unable to efficiently surmount multiple steps and so dissipate more efficiently (Altan-Bonnet & Germain,

2005). An important corollary of these conclusions is that, at least for the ones we have investigated, the opposing reactions within the signaling network that seek to re-establish the basal level of signaling must be efficient and potent, with continuous signaling flux required to overcome them.

Previous studies have investigated how transient inhibition of TCR signaling affects the level of $Ca^{2+}$ fluxing and the rate of its decline on signal disruption (Valitutti *et al*, 1995; Varma *et al*, 2006; Yousefi *et al*, 2019). These reports found a strong dependence on proximal TCR signaling to maintain the increased level of $Ca^{2+}$ ions in agreement with our findings. However, we measured a far more rapid decrease in this readout (Fig 2L), which we attribute to the way we disrupt receptor signaling being more direct and efficient. The rapid decrease in phosphorylated ERK on disruption of signaling that we observed (Fig 3E) has also been found in another recent optogenetic study in NIH3T3 cells (Bugaj *et al*, 2018), suggesting that the results we have found are likely to be more generally applicable beyond T-cell signaling. Our finding that the active state of the transcription factor FOS, like ERK, required continuous proximal signaling implies that only their downstream output of increased gene expression can constitute a significant persistent state of previous signaling for these TFs. We do not imply that other persistent states cannot exist though, such as migration or cytotoxic activity through other mechanisms. This means that the encoding of signaling dynamics through TFs should be at this level too, and previous studies have provided evidence for this result (Murphy *et al*, 2002; Locasale, 2007; Clark *et al*, 2011; Marangoni *et al*, 2013).

There have been several other studies employing optogenetics to investigate how signaling dynamics influence downstream cellular activation, which has led to some exciting new results (Toettcher *et al*, 2013; Graziano *et al*, 2017; Wilson *et al*, 2017; Bugaj *et al*, 2018). These studies invariably used the light-mediated translocation of a constitutively active enzyme to the plasma membrane as the means to control downstream signaling. While very effective, this approach "short-circuits" signaling from the upstream receptor, potentially bypassing key parts of the network and removing any spatio-temporal information that might be encoded in physiological receptor activation within cell conjugates. In our approach, light controls the very initiation of receptor signal transduction without altering the network architecture, with receptor activation occurring within the physiological context of a T-cell/presenting cell conjugate interface. We believe this approach provides the most realistic control over receptor activation, without disrupting signaling from other cell surface receptors that might modulate the TCR input, such as costimulation through CD28 engagement. We have also investigated the endogenous signaling components rather than over-expressing protein sensors, which should most faithfully report the dynamics of the network.

The OptoCAR we have developed in this work contains three ITAM signaling motifs. While capable of initiating very efficient downstream signaling, it cannot be expected to fully replicate all the aspects of the complete TCR complex. We have previously shown that the orthogonal receptor/ligand pair used in the OptoCAR can drive equivalent segregation of the phosphatase CD45 that is thought to initiate receptor signaling as observed for the TCR (James & Vale, 2012) as well driving downstream signaling (James, 2018). We have also shown that the increased number of ITAMs present on the TCR allows it to be highly efficient at transducing ligand binding into a signal even at low receptor occupancy (James, 2018). However, for all the experiments performed in this work we have worked in a high occupancy regime by saturating with the dimerizer, so we expect that the OptoCAR, like similar CAR constructs, provides a broadly equivalent stimulus to that expected from the TCR itself.

We have used the Jurkat cell line for most of our assays, which is a well-used model for T-cell studies (Abraham & Weiss, 2004) but cannot entirely replicate the functions of primary $CD4^+$ T cells (Bartelt *et al*, 2009). However, the known mutation in Jurkat cells in PI3K signaling due to PTEN inactivation (Shan *et al*, 2000) would not be expected to have any direct effect on the dynamics of the signaling pathways investigated in this study. We are therefore confident that the conclusions we have drawn from our datasets are likely to generalize to other receptor signaling networks that depend on the same transduction pathways.

In summary, we have developed an optically controllable antigen receptor to quantify the temporal window over which T cells can integrate TCR signals between successive APC interactions, and reinforce the view that sustained proximal signaling is required for potent cell activation.

# Materials and Methods

### Reagents and Tools table

| Reagent or Resource | Source | Identifier |
|---|---|---|
| **Antibodies** | | |
| Mouse anti-human CD86 (Alexa Fluor 647) | BioLegend | Cat#: 305416 |
| Human TruStain FcX (FcR blocking) | BioLegend | Cat#: 422302 |
| Phospho-Erk1/2 ($Thr^{202}$/$Tyr^{204}$) mAb (Alexa Fluor 488) | CST | Cat#: 4374S |
| Rabbit anti-phospho-FOS (Ser32) mAb | CST | Cat#: 5348S |
| Rabbit anti-human FOS mAb | CST | Cat#: 2250S |
| Mouse anti-human CD69 (Alexa Fluor 647) | BioLegend | Cat#: 310918 |
| Mouse anti-human CD3 (Alexa Fluor 647) | BioLegend | Cat#: 300322 |

**Reagents and Tools table** (continued)

| Reagent or Resource | Source | Identifier |
|---|---|---|
| Mouse anti-human CD20 (Brilliant Violet 605) | BioLegend | Cat#: 302334 |
| Mouse anti-human CD137 (Brilliant Violet 421) | BioLegend | Cat#: 309820 |
| **Bacterial and virus strains** | | |
| *Escherichia coli* DH5alpha | Thermo Fisher | Cat#: 18265017 |
| **Chemicals, peptides, and recombinant proteins** | | |
| A/C dimerizer (AP21967) | Takara | Cat#: 635057 |
| Indo-1 LR (AM) | TEFLabs | Cat#: TEF0145 |
| Pluronic F-127 | Sigma-Aldrich | Cat#: P2443 |
| Lenti-X concentrator | Takara | Cat#: 631231 |
| Perm buffer III | BD | Cat#: 558050 |
| **Critical commercial assays** | | |
| RNeasy Plus Mini Kit | Qiagen | Cat#: 74134 |
| QIAshredder | Qiagen | Cat#: 79654 |
| SuperScript IV VILO Master Mix with ezDNase | Thermo Fisher | Cat#: 11756050 |
| TaqMan Advanced Fast Master Mix | Thermo Fisher | Cat#: 13428456 |
| *GAPDH* TaqMan assay | Thermo Fisher | Hs02786624_g1 |
| *PGK1* TaqMan assay | Thermo Fisher | Hs00943178_g1 |
| *CD69* TaqMan assay | Thermo Fisher | Hs01119267_g1 |
| *FOS* TaqMan assay | Thermo Fisher | Hs00934033_m1 |
| *IL2* TaqMan assay | Thermo Fisher | Hs00174114_m1 |
| *CXCL8* (*IL8*) TaqMan assay | Thermo Fisher | Hs00174103_m1 |
| *TNFRSF9* (*CD137*) TaqMan assay | Thermo Fisher | Hs00155512_m1 |
| **Experimental models: cell lines** | | |
| HEK-293T: human adherent cell line | ATCC | Cat#: CRL-11268 |
| J.NFAT: derived from human Jurkat T-cell leukemia line | (James, 2018) | N/A |
| Raji-FRB$^{Ex}$: derived from human B-cell lymphoma line | (James & Vale, 2012) | N/A |
| **Oligonucleotides** | | |
| For primer sequences, see Appendix Fig S3 | This work | N/A |
| **Recombinant DNA** | | |
| pCMVΔR8.91 | (Zufferey *et al*, 1997) | N/A |
| pMD2.G | Gift from Didier Trono | Addgene#: 12259 |
| pHR-OptoCAR-mScarlet | This work | Addgene#: 170463 |
| pHR-iRFP670-CaaX | This work | Addgene#: 170464 |
| pHEFI-OptoCAR$^{CD19}$-mScarlet | This work | Addgene#: 170465 |
| **Software and algorithms** | | |
| Fiji (ImageJ): image analysis | (Schindelin, Arganda-Carreras *et al*, 2012) | https://fiji.sc/ |
| MATLAB R2019a: data processing | MathWorks | N/A |
| FlowJo V10.2: data processing | FlowJo LLC | N/A |
| Excel 365 | Microsoft | N/A |
| Illustrator 2020 | Adobe | N/A |

## Methods and Protocols

### Cell culture

HEK-293T cells were grown in Dulbecco's modified Eagle's medium supplemented with 10% fetal bovine serum (FBS) and antibiotics. Jurkat (J.NFAT) cells (James, 2018) and Raji-FRB$^{Ex}$ (James & Vale, 2012) cells were grown in RPMI 1640 medium, supplemented with 10% FBS, 10 mM HEPES, and antibiotics. Primary T cells were isolated and grown as described previously (James, 2018). All cell cultures were maintained at 37°C with 5% $CO_2$ in a fully humidified incubator.

### OptoCAR vector construction and lentiviral transduction

To create the optically controllable chimeric antigen receptor termed OptoCAR in the main text (Appendix Fig S3A and B), we first amplified the LOV2 light-sensing domain from *Avena sativa* phototropin 1 using primers pr1/pr2 (Appendix Fig S3C). We then fused this part to the extracellular and transmembrane regions of our previous chemically inducible CAR, FKBP-CD86-CD3ζ (James, 2018), by inserting it as a *SpeI/NotI* fragment, replacing the ζ-chain sequence. This FKBP-CD86-LOV2 construct was then excised as a *MluI/NotI* fragment and inserted into a vector (pHR-1G4) that can accommodate two genes separated by the P2A ribosomal skip sequence (James, 2018). The intracellular sequence of the bipartite OptoCAR was gene synthesized (LCK myristoylation sequence, Zdk, ζ-chain, and mScarlet fluorophore; see Appendix Fig S3A) and fused to P2A and FKBP-CD86-LOV2 by overlap extension PCR using primers pr3-pr5 and pr1 (Appendix Fig S3C). This process also removed a *SpeI* site at the 3′ end of mScarlet to make it unique at the 5′ end of LOV2. The single contiguous open reading frame was inserted into pHR-1G4 as an *AsiSI/NotI* fragment (Appendix Fig S3A).

To create LOV2 mutants for some of the OptoCAR experiments (C450G [TGC->GGC], I539E [ATT->GAA], V416L [GTC->CTC]), we used site-directed mutagenesis and cloned them as *SpeI/NotI* fragments into the OptoCAR vector. All vector sequences were confirmed by Sanger sequencing. For the variant of the OptoCAR that used a cytoplasmic intracellular sequence, the complete section of Zdk-CD3ζ-mScarlet was simply amplified without the myristoylation region (but now including a Kozak sequence) using pr6/pr7 (Appendix Fig S3C) and replaced the equivalent region in the normal OptoCAR vector as a *AsiSI/BamHI* fragment.

The OptoCAR sequence was introduced into J.NFAT cells by lentiviral transduction in an equivalent manner described previously (James, 2018). This invariably led to cells that were uniformly positive and had a tight distribution of expression such that there was no need to sort or clone OptoCAR-T cells.

For the OptoCAR[CD19] construct (Appendix Fig S3D and E), the extracellular and transmembrane region of anti-CD19 CAR was amplified as a *MluI/SpeI* fragment and replace the equivalent section in the OptoCAR vector.

### Flow cytometry

Cell samples to be analyzed by flow cytometry in standard 12 × 75 mm tubes were centrifuged at 800 $g$ for 3 min, with pellets flick resuspended. Where required, primary antibodies were used at a working concentration of ~10 μg/ml, diluted in flow wash buffer [2.5% (v/v) FBS, 0.1% (w/v), NaN$_3$ in PBS, pH 7.4], and incubated on ice for at least 30 min, with regular agitation. For Raji B cells, endogenous Fc receptors were pre-blocked with Human TruStain FcX for 10 min at room temperature. Samples were washed with 3 ml flow wash buffer and centrifuged as before. When the primary antibody was fluorescently conjugated, the pellet was resuspended and fixed in flow fix buffer [1.6% (v/v) formaldehyde, 2% (w/v) glucose, 0.1% (w/v) NaN$_3$ in PBS, pH 7.4]. When a secondary labeled antibody was required, an equivalent staining protocol was used, with the secondary antibody at 20 μg/ml, prior to washing and fixing.

For staining in 96-well round-bottomed plates, samples were transferred into the plate using a multichannel pipette, ensuring cells were properly mixed by gentle pipetting up and down. The plate was then centrifuged at 800 $g$ for 3 min, and the supernatant was discarded. Cell pellets were resuspended in the residual media by pulsed vortexing. When required, primary antibodies at ~10 μg/ml, diluted in flow wash buffer (100 μl/well), were added to all wells and incubated on ice for at least 30 min. Wells were then washed by adding 150 μl FACS wash, and the plate was centrifuged at 800 $g$ for 3 min. The supernatant was discarded, and the plate was vortexed to resuspend the cell pellet in the residual solution. A second wash with 200 μl was performed similarly. Finally, 200 μl FACS fix buffer was then pipetted into the wells and the plate was stored at 4°C until analyzed.

The flow cytometer was set up to ensure the most appropriate dynamic range between the negative and positive controls for each fluorescence channel, and at least 30,000 gated cells were collected for each analysis. All data were collected on either a BD LSRII or Fortessa flow cytometer, with the latter instrument having an HTS unit to collect data from the 96-well plates.

### Measurement of OptoCAR dissociation kinetics by microscopy

To quantify the rates at which the bipartite OptoCAR construct dissociated on illumination and reformed on return to the dark condition, we transiently transfected HEK-293T cells with the OptoCAR variant that had a cytoplasmic intracellular part, in combination with prenylated iRFP670 fluorophore (iRFP-CaaX) to demark the plasma membrane. Transfected cells were then adhered to fibronectin-coated imaging dishes and placed on a spinning disk confocal microscope that was maintained at 37°C and 5% CO$_2$. A 100× oil-immersed objective was used to image the OptoCAR construct (mScarlet: 561 nm Ex; 607 ± 18 nm Em) and iRFP-CaaX (640 nm Ex; 708 ± 37 nm Em) sequentially at 1 Hz. These fluorophores were utilized as imaging them would not have any influence on the conformational state of LOV2. To simultaneously image and illuminate cells, we placed a blue (450 ± 25 nm) filter in the condenser of the transmitted light path using a custom-designed holder and closed down the field diaphragm so only the cells in the field of view were illuminated. After 25 s of imaging, the transmitted white light LED was switched on to illuminate the sample region in focus for a further 25 s before turning the LED off. Imaging continued for a further 100 s.

Image stacks were subsequently analyzed in MATLAB. The two channels were first registered to optimally align the stacks before the cell regions of interest (background, cytoplasm, whole cell) were user-defined and both channels were background subtracted and bleach corrected. The iRFP-CaaX channel was then thresholded and Gaussian bandpass-filtered to define a binary representation of the plasma membrane for each frame. This mask was then used to calculate the mean intensity of the OptoCAR intracellular section compared with the mean intensity from the cytoplasm for each frame. For the main figure, the datasets were normalized so that the mean ratio of membrane fluorescence was scaled to unity and the minimum value after illumination to zero. However, to extract the rates of light-induced dissociation and re-association of the OptoCAR in the dark state, single exponential curves were fit to both the respective phases of each individual cell time course without normalization.

### Formation of cell conjugates

Many of the assays required conjugating J.NFAT cells expressing the OptoCAR with Raji-FRB[Ex] cells in the absence of the dimerizer,

prior to starting the experiment itself. Generally, a specified number of OptoCAR-T cells and Raji-FRB$^{Ex}$ cells were first centrifuged separately at 800 $g$ for 3 min and the cell pellet flick resuspended in residual medium. A small volume of fresh medium was then added to the OptoCAR-T cells before combining with the Raji-FRB$^{Ex}$ cells, such that the cells were at high density at a 1:1.5–2 ratio. Usually, this cell mixture was then aliquoted into separate 1.5-ml Eppendorf tubes and centrifuged at 800 $g$ for 1 min to drive the cells into close apposition and incubated at 37°C for between 10 and 30 min. After this incubation, the cell pellets in each tube were gently resuspended by pipetting up and down, fresh medium was added to increase the volume and then used directly in the required downstream application.

### Measurement of Ca²⁺ flux by flow cytometry

OptoCAR-T cells were labeled with the fluorescent $Ca^{2+}$ indicator, Indo-1 AM by resuspending $5 \times 10^5$ cells in 500 µl serum-free medium containing Indo-1 at a final concentration of 5 µg/ml, supplemented with 0.1% Pluronic F-127. The cells were loaded with Indo-1 at 37°C for 30 min. Labeled cells were then washed using 3 ml PBS (with $Ca^{2+}$ and $Mg^{2+}$) to remove any excess Indo-1 and flick resuspended in residual volume. Conjugates of Indo-1 loaded cells and Raji-FRB$^{Ex}$ cells were prepared as described above and added to standard flow cytometry tubes ready for sample acquisition.

To optically modulate the conjugated cells while measuring the intracellular $[Ca^{2+}]$ on the flow cytometer, we custom-built a device that could illuminate the sample tube with a 455 nm wavelength LED (royal blue) while inserted in the cytometer. We also included a thermostatically controlled heating block around the tube so that the desired temperature could be maintained during data acquisition. Conjugated cells were run on the flow cytometer for 60 s to define a baseline ratiometric signal of Indo-1 fluorescence (405 ± 15 nm / 485 ± 13 nm) excited at 355 nm. The sample tube was then removed and 2 µM dimerizer added to initiate signaling within the conjugates, which were gated on as iRFP713$^+$/mTagBFP$^+$ doublet events. Events were collected for varying periods depending on the type of experiment, and when required, the LED illuminator was switched on manually at the appropriate time.

For data analysis, cytometry files were first gated in FlowJo before exporting CSV data files that were further processed in MATLAB. The Indo-1 ratio was calculated after background subtraction from the individual channels, and then, a 100-bin histogram formed for data points binned over every 2 s. Density plots were then formed by stacking these histograms vertically as shown in the main figures. For fitting the $Ca^{2+}$ flux dynamics, the mean of the Indo-1 ratio at each time interval was first calculated and plotted. The light-induced decay of this signal was fit to the survival function of a gamma distribution over a range of shape parameters, which corresponds to the number of mass action steps in a linear reaction scheme.

### Phospho-ERK dynamics

To follow the dynamics of ERK phosphorylation, $0.5 \times 10^6$ J.NFAT cells expressing the OptoCAR were conjugated with $1.5 \times 10^6$ Raji-FRB$^{Ex}$ cells as described above. Prior to the initiation of receptor activation, a sample was taken to define the 0-min time point. After this, the dimerizer was added at a final concentration of 2 µM and cells were incubated at 37°C. After the required time, 100 µl of cell suspension was removed and immediately fixed in 500 µl 4%

formaldehyde prewarmed to 37°C for 10 min. Samples were then kept on ice until time points had been acquired so that they could be further processed. Fixed samples were washed with 3 ml flow wash buffer and centrifuged at 800 $g$ for 3 min. The resuspended pellet was then vortexed, while 500 µl perm buffer III (cooled to −20°C) was added dropwise. After this, samples were left on ice for 30 min and then washed twice with 3 ml flow wash buffer. The resuspended pellet was then incubated with the phospho-ERK1/2 (Thr$^{202}$/Tyr$^{204}$) antibody directly conjugated with Alexa Fluor 488 for 45 min at room temperature, with regular agitation. Stained cells were then washed with 3 ml flow wash buffer and fixed with 500 µl flow fix buffer before data acquisition by flow cytometry. Despite methanol fixation, the fluorescence from the iRFP713$^+$ J.NFAT cells and mTagBFP$^+$ Raji B cells was still readily detectable, allowing us to gate exclusively on cell conjugates for the subsequent analysis of phospho-ERK intensity.

### Construction and calibration of the optoPlate illumination device

For a significant number of experiments in this work, we made use of an open-source illumination device, termed an "optoPlate" that provides a way to independently illuminate all wells of a 96-well plate (Bugaj & Lim, 2019). We followed the authors' protocol to build our own optoPlate using blue (470 nm) LEDs in both positions under each well, with the following modifications. We increased the depth of the plate holder to 18mm and used 5-mm-thick perspex squares as the diffusers in each position to improve the uniformity of illumination over the entire well. We also used a Raspberry Pi computer to communicate with the Arduino Micro onboard the optoPlate so that update commands for the intensity of each well could more easily be modulated. The Raspberry Pi used a simple CSV file written using MATLAB to send the intensity values for each well at an arbitrary refresh rate, normally every 10 s. This also allowed us to check on the status of the experiment by remote connection to the optoPlate while it was in the incubator.

Before any experiments were performed, the wells of the optoPlate were calibrated such that a defined pulse-width modulation (PWM) value applied to the LEDS of one well to control its intensity would be equivalent across the entire plate. We first imaged the optoPlate (without 96-well plate) using a gel doc system at a range of PWM values, performed separately for the LED well pairs. We then used a MATLAB script to threshold the images to create well masks so that the mean intensity for each well could be calculated. We confirmed that all wells showed linearity of light intensity with PWM value before calculating calibration values so that the "dimmest" well defined the maximum output and all other wells were proportionately scaled down. These calibration values were hardcoded onto the Arduino so that no calibration of the experimental program was required. We confirmed that this calibration process led to even illumination across the optoPlate, with a standard error of 0.4%.

### FOS transcription factor dynamics

OptoCAR-T cells were conjugated as described above and dimerizer was added to 2 µM to initiate receptor signaling. For sample consistency, a stock of activated conjugates was prepared before adding 2 µM dimerizer and aliquoting into 24 wells of the optoPlate. We used Greiner µClear flat-bottomed plates for all experiments, which provided excellent light penetration and no spillover between wells.

The first sample was then immediately taken to assay FOS levels at onset of activation. At the appropriate time point, all cells were removed from the required well (100 µl) and pipetted into an Eppendorf tube containing ice-cold 800 µl PBS. The tube was then centrifuged at 2,000 $g$ for 30 s, supernatant was discarded, and 100 µl RIPA buffer was added to the pellet and gently mixed. The lysed sample was incubated in ice for at least 30 min. When sample illumination was required, half of the optoPlate was switched on so that both "light" and "dark" samples could be collected simultaneously. Once all samples had been collected, the lysed cells were centrifuged at ~16,000 $g$ for 10 min at 4°C. The supernatant (90 µl) was then pipetted into PCR tubes containing 42 µl of reduced LDS sample buffer and heated in a thermal cycler at 70°C for 10 min.

To detect total FOS and phosphorylated FOS protein levels, 15 µl of all samples was loaded onto two 26-well 8% Bis–Tris polyacrylamide gels and subjected to gel electrophoresis with a pre-stained protein ladder (BlueElf) in the end wells of the gel. Gels were then transferred to a nitrocellulose membrane using the iBlot2 dry blotting system. Blots were air-dried for 5 min before using No-Stain Protein Labelling Reagent to provide total protein normalization (TPN) loading control, following the manufacturer's protocol. The blots were then immersed in blocking buffer (2.5% milk powder in Tris-buffered saline (TBS)) for 60 min. Primary antibody solutions were prepared in 10 ml blocking buffer supplemented with 0.1% Tween-20 (TBS-T/milk) using the FOS antibodies described in the Key Resources Table at manufacturer's recommended dilution. Blots were incubated with antibody solutions overnight at 4°C on a rocking platform. After aspirating the primary antibody, blots were washed four times with TBS-T/milk before adding 10 ml secondary antibody solution containing CF790 anti-rabbit IgG(H + L) antibody at 1:10,000 dilution in TBS-T/milk. Blots were incubated for 60 min at room temperature on a rocking platform before being washed four times with TBS-T buffer. Blots were left in TBS and immediately imaged using the camera-based Azure 600 fluorescence detector.

To quantify the Western blot images, a line profile of each lane on the blots was measured using FIJI for both the TPN and FOS fluorescence channels and exported into MATLAB for further analysis. The band profiles were aligned through an automated script that used the TPN data to calculate the alignments that were applied to the FOS dataset. Background intensity was subtracted before numerical integration of the total peak intensities was calculated. For the TPN data, almost all the lane intensity was included in this integration to give the most reliable measure of protein loading. To calculate the "active" fraction of FOS when required, a binary gate was applied to the histogram of the lane profile to define the higher molecular weight part of profile. This dataset was normalized between the 0- and 15-min time points.

### Reverse transcription quantitative PCR (RT-qPCR) to measure mRNA levels

OptoCAR-T cells and Raji cells were conjugated similar to other assays such that each well contained $3.2 \times 10^5$ OptoCAR-T cells conjugated with $5.6 \times 10^5$ Raji-FRB$^{Ex}$. A pooled stock of activated conjugates was prepared before adding 2 µM dimerizer to initiate the receptor signaling and then aliquoting cell conjugates into 26 wells of Greiner µClear flat-bottomed 96-well plate. Eleven wells were illuminated after 3 h, with the remainder of wells kept in the

dark for the entire assay. A volume of 100 µl was removed for each sample and put into 1.5-ml Eppendorf tube containing 800 µl PBS. The tube was centrifuged at 500 $g$ for 30 s, and the supernatant was discarded before placing tube in a pre-cooled metal block in dry ice to rapidly freeze the cell pellets. Other samples were taken out in same manner at specified time points.

Frozen cell pellets were thawed on ice, and then total RNA was extracted using RNeasy Plus Mini Kit (Qiagen) following manufacturer's protocol, with QIAshredder columns used to homogenize samples and reduce viscosity. The total RNA concentration was measured using NanoDrop Spectrophotometer (Geneflow) at 260 nm, and RNA quality was assessed by measuring 260/280 nm absorbance ratio. cDNA was reverse-transcribed from 2 µg of RNA using SuperScript IV VILO Master Mix with ezDNase Enzyme (Invitrogen) following manufacturer's protocol; a "No-RT" control sample was always prepared in conjunction.

The relative mRNA levels of defined genes were analyzed by quantitative real-time PCR, with *GAPDH* and *PGK1* used as reference housekeeping genes. The 20 µl PCR mix was prepared with 10 µl TaqMan Advanced Fast Master Mix, 1 µl TaqMan Assay for each gene, 7 µl nuclease-free water, and 2 µl cDNA sample. The reaction mixtures were pipetted into 96-well PCR plates and sealed with optical Cap strips. The plate was vortexed to mix the reagents and then briefly centrifuged to bring the reaction mix to the bottom of each tube and eliminate air bubbles. The plate was then loaded onto an Agilent Mx3005P Real-Time PCR System and run using cycling conditions as follows: UNG(uracil-N-glycosylase) incubation at 50°C for 2 min, polymerase activation at 95°C for 2 min followed by 40 PCR cycles of denaturation at 95°C for 3 s, and annealing/extension at 60°C for 30 s.

The cycle threshold (Ct) value for each PCR was extracted from the fluorescent traces and converted to a relative quantity (RQ) by taking the difference between each time point and the initial value at the start of the experiment. We confirmed that the efficiencies of each TaqMan assay were ~100% over the range of extracted Ct values. The RQ values were then corrected (cRQ) for total input using the geometric mean of the paired RQ values for *GAPDH* and *PGK1* housekeeping genes. These values were then normalized over [0,1] using the mean cRQ values between 180 and 240 min for each dataset and this is what is plotted in the main figure. All experiments are represented as the mean of three biological replicates using single or double technical replicates.

### Measurement of downstream OptoCAR output

To provide more robust control over downstream signaling, we used the V416L variant of LOV2 (Wang *et al*, 2016) in the OptoCAR, which has a slower dark reversion rate and requires lower light intensity to maintain signaling quiescence. This had the benefit of keeping the cell conjugates healthier over the 24-h illumination period by minimizing any phototoxicity effect. OptoCAR-T cells were conjugated with ligand-presenting cells as described above. Generally, the light intensity used to disrupt signaling was 250 µW/cm$^2$. For reproducibility, a large stock of cell conjugates was prepared using $8 \times 10^6$ OptoCAR-T cells and $15 \times 10^6$ Raji-FRB$^{Ex}$ cells, before adding 2 µM dimerizer and aliquoting 200 µl of conjugates into all wells of a 96-well plate, which corresponded to $8 \times 10^4$ OptoCAR-T cells and $1.5 \times 10^5$ Raji-FRB$^{Ex}$ cells per well. We used Greiner µClear flat-bottomed plates for all experiments, which provided

excellent light penetration and no spillover between wells. The optoPlate was then placed in a 37°C/5% $CO_2$ incubator and run for 24 h with individual well illumination profiles appropriately programmed depending on the required experiment. Signaling ("dark") pulses were 15, 30, 60, 72, 90, or 120 min in duration and always summed to 6 h, e.g., twenty-four 15-min pulses or four 90-min pulses. These signaling pulses were separated by different time intervals of the light-induced "off" state. The range of these light intervals was calculated similarly to a pulse-frequency modulation experiment by varying the duty cycle between 100 and 25%. For example, for 30-min pulses, the light intervals used ranged from 0 min (6 h of continuous) to 90 min. Duplicates wells of constant dark, constant light, and 6 h continuous dark were included for all experiments. The illumination profiles for each experiment were randomized across the optoPlate to remove any potential positional effect that might have biased the results. On completion of the experiment, samples were stained for CD69 surface expression in 96-well round-bottom plates and analyzed by flow cytometry as described above.

Experimental datasets were analyzed using FlowJo to gate on OptoCAR-T cells and quantify the geometric mean of the GFP and CD69 (Alexa Fluor 647) expression distributions. These values were then normalized between 6 h of continuous signaling and the constant light control and plotted against the inter-pulse interval.

### Cellular response to varying signaling potency

To titrate the signaling "strength" within OptoCAR-T cells, the optoPlate was used to continuously illuminate cell conjugates over a range of intensities, with the experimental set up in an equivalent manner to the previous sections. The gated sample data from FlowJo were then imported into MATLAB for further analysis. The distributions of fluorescence intensity from both NFAT-mediated GFP expression and CD69 upregulation were constructed for each well before combining vertically to build a density plot with downstream output expression against illumination intensity.

### Microscopy of OptoCAR-T cell conjugate activation

To quantify expression of NFAT-mediated GFP expression over time from individual cell conjugates, we used live-cell spinning disk confocal microscopy. OptoCAR-T cell conjugates were prepared as described above but using the LOV2 C450G mutation so that the OptoCAR remained in the signaling-competent state even during imaging at wavelengths < 500 nm that would normally disrupt it. Conjugated cells were pipetted into a fibronectin-coated imaging dish and placed on the microscope maintained at 37°C and 5% CO2. A 100× oil-immersed objective was used to image the J.NFAT cells (iRFP713: 640 nm Ex; 708 ± 37 nm Em), Raji-FRB[Ex] cells (mTagBFP: 405 Ex; 460 ± 30 nm Em), OptoCAR construct (mScarlet: 561 nm Ex; 607 ± 18 nm Em), and GFP (488 Ex; 525 ± 25 nm Em) every 10 min for 24 h. In order to image a significant number of conjugates at this high resolution, we imaged a 12 × 12 grid of sample regions (512 × 512 px/region) with 10% overlap and then stitched the separate stacks together using "Grid Stitching" in FIJI. This stitched image stack was then split into 9 sub-stacks to facilitate analysis and five of these regions maintained enough conjugates over 24 h to quantify the mean GFP intensity over time.

### Measurement of OptoCAR[CD19] activation

Human primary CD4[+] T cells were transduced with the anti-CD19 OptoCAR (OptoCAR[CD19]) construct as previously described (James, 2018). The OptoCAR[CD19]-T cells were conjugated with Raji B cells as for the OptoCAR experiments above (but without the dimerizer addition) and various illumination profiles were patterned onto the cells over 24 h using the optoPlate. On completion of the experiment, samples were stained for CD3 and CD20 to differentiate the two cell types by flow cytometry, as well as CD137 to quantify T-cell activation. All OptoCAR[CD19]-positive T cells (as gated by mScarlet fluorescence) were found to remain conjugated with B cells so this population was quantified for CD137 expression. These values were then normalized between 6 h of continuous signaling and the constant light control and plotted against the inter-pulse interval.

## Data availability

Source data for main figures can be found in Dataset EV1. Plasmid sequences are available through the Addgene repository.

**Expanded View** for this article is available online.

## Acknowledgements

We are very grateful to the mechanical and electronic workshops at the MRC-LMB for assistance with construction of the illumination devices used in this study, and to Klaus Hahn and Lukasz Bugaj for sharing reagents that were invaluable to this work. We thank members of the James Lab for commenting on the manuscript. This work was supported by the Wellcome Trust (Grants 099966/Z/12/Z to J.R.J. and 102195/Z/13/Z to M.J.H) and Warwick Medical School. M.F acknowledges funding from the University of Warwick, the EPSRC & BBSRC Centre for Doctoral Training in Synthetic Biology (Grant EP/L016494/1). We acknowledge equipment access, training, and support made available by the Research Technology Facility of the Warwick Integrative Synthetic Biology Centre (WISB), which received funding from EPSRC and BBSRC (BB/M017982/1).

## Author contributions

MJH and JRJ conceptualized the study; MJH, MF, and JRJ designed methodology; MJH, MF, and JRJ investigated the data; JRJ wrote the original draft; MJH, MF, and JRJ wrote, reviewed, and edited the manuscript; JRJ visualized the data; JRJ supervised the data; and JRJ acquired funding.

## Conflict of interest

The authors declare that they have no conflict of interest.

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
