## [Review Process File · Molecular Systems Biology]

Quantifying persistence in the T cell signaling network using an optically controllable antigen receptor

Michael Harris, Muna Fuyal, and John James

DOI: [10.15252/msb.202010091](https://doi.org/10.15252/msb.202010091)

Corresponding author(s): John James (john.james@warwick.ac.uk)

Review Timeline:

Submission Date:	3rd Nov 20
Editorial Decision:	7th Dec 20
Revision Received:	12th Mar 21
Editorial Decision:	15th Apr 21
Revision Received:	21st Apr 21
Accepted:	22nd Apr 21

Editor: Maria Polychronidou

Transaction Report:

Thank you again for submitting your work to Molecular Systems Biology. We have now heard back from the three referees who agreed to evaluate your study. Overall, the reviewers acknowledge that the study seems potentially interesting. They raise however a series of concerns, which we would ask you to address in a revision.

I think that the reviewers' recommendations are rather clear and relatively straightforward to address, and I therefore see no need to repeat the comments listed below. All three reviewers provide constructive suggestions for improving the study. Please let me know in case you would like to discuss in further detail any of the issues raised. All issues raised by the referees would need to be satisfactorily addressed.

On a more editorial level, we would ask you to address the following points.

REFeree REPORTS

Reviewer #1:

Summary of the paper -

This well-written manuscript presents an interesting series of results dissecting receptor-proximal

signal persistence, timescales, and additivity, in a system engineered to provide dual chemical and optogenetic control of T-cell antigen receptor signaling.

The contributions are well-laid out, with a clear logical sequence as follows -

1. A description of the engineered OptoCAR where light-controlled separable domains are inserted between the membrane-bound receptor and its intracellular signaling domains, accompanied by a validation of its expected behaviour in non-immune cells and characterization of light-reversible association-dissociation kinetics.
2. Investigation of successively distal constituents of the signaling network downstream of the receptor, showing the elongation of signal dissipation timescales further down the signaling cascade :
 - (i) Intracellular calcium fluxes, which dissipate rapidly within seconds of receptor deactivation
 - (ii) Receptor-proximal cytosolic signaling by ERK, followed by nuclear localization of FOS/JUN, which preserve memory on the short timescale of a few minutes
 - (iii) De novo gene expression of NFAT-driven GFP and CD69, which translates these signals into outputs, with a persistence determined by mRNA half-life scales of up to 30 minutes
 - (iv) Further comparison with mRNA expression levels of other response genes, showing effects of temporal coupling between signaling and transcription
3. A comparison between continuous and pulsatile signaling at distal stages having longer persistence, showing the downstream effects of interaction between pulses.
4. Application of such pulsatile signaling to a clinically-relevant CAR, where the spacing between signals sustains activity over time by allowing relaxation of the signaling network, thus avoiding saturation and negative feedback loops.

Some of the major strengths of the paper -

1. Although the paper covered diverse technical and biological aspects - such as the biology of TCR receptor signaling itself and its downstream components, the design of the optogenetic system and associated experimental platforms for tuned illumination, and a practical application in the context of a clinically relevant CAR - the background for each aspect was laid out well enough for any reader not expert in a particular area to still be able to place results in context and appreciate the work as a whole.
2. The requisite controls for all experiments have been considered thoroughly, and were comprehensively and convincingly presented in the supplements to bolster all the major claims.
3. The paper appears technically very sound, with detailed discussion of the observations, clear understandable figures, and well-supported conclusions.

Questions, comments, and suggestions for the manuscript -

Overall, the technical aspects and results are mostly well-presented. The discussion section may have some more potential for depth, since conclusions in this work can be considered in some more interesting directions.

Results section -

- (1) It might be easier for a reader to first see Figure 6 and the section titled "Gene transcription is rapidly abolished after disruption of receptor signaling", prior to Figure 5 and the section "Short-term signal persistence can be directly observed in gene expression output".
 - Figure 6 and the mRNA expression time-courses serve as a nice extension to the work presented on calcium signaling, FOS, and ERK, with transcription being the next logical step in the cascade, and the half life being extended from ~10min to ~30min. This also showcases mRNA production as

a step that extends the "memory" in the system for a while longer, facilitating the chaining of signals separated by longer period of time, which have already decohered at earlier nodes in the network. With this background in place first, it becomes easier to grasp why integration is possible, even when the pulses are individually undetectable.

- Figure 5 extends that by one step, to the outputs of transcription and translation - that is, observable protein levels of GFP and CD69, which may be considered the final output of this cascade. Having the mRNA information in place before looking at protein expression is just easier to follow, and also consistent with the layout of this manuscript, presenting successively distal processes from the receptor.

- Figure 5 is also a better segue to Figure 7, connecting the concepts through pulsatile signaling.

(2) The section on "Short-term signal persistence can be directly observed in gene expression output" could be much more clearly presented and explained to the reader. There are some aspects that are really not clear, or possibly very interesting but insufficiently addressed.

- Subplot 5E would ideally come first in this figure, since it shows the expression kinetics over time, which is the baseline for everything else discussed here - saturation, pulses, etc.

- For Figures 5A-D and Lines 308-311, it appears that the total signal duration ranges up to 24 hours total, but only end-point expression is measured, uniformly at the end of 24 hours in all cases? This would be helpful to specify very explicitly in the text, with a clear explanation of why this design was chosen (i.e. why 24-hours).

- In Figures 5A-E, and Lines 315-318, why is there a broad claim that the system is not saturating? This seems clearly contradicted by Fig 5E, where the expression curve is obviously reaching saturation for signal durations approaching 24 hours, and Figs 5A,C where the population appears to show saturation in the number of responders (and their intensity) beyond about 10 hours. Also, Figs 5B,D which do not show saturation are on a different X-axis scale of only 6 hours - why is this also not extended to 24 hours? It's true that there's no saturation within this 6-hour regime (which is probably very relevant for the next steps in Figs 5F,G, and maybe why this is highlighted explicitly in the text), but there's clearly saturation over a 24-hour signaling period! This should be properly disentangled, to be consistent throughout and less confusing.

- In Lines 321/339, please quantify what is meant by a detectable pulse - is it a 5% increase in expression level above the baseline, or just the first increase of any magnitude? How is signal distinguished from noise at this low level, and how does the 30-minute level compare with the variations observed in these measurements?

- In Figures 5A-D, when the end-point was measured, it was hard to tell if this is just at the end of the dark signaling period, or the end-point was after a full 24 hours? If it's the former, then the end-point is related to the signal duration. But if it's the latter, then the outcome is a consequence of both activation in the dark period and dissipation during the light period, both of which have varying durations that cannot be decoupled. In this case, it's hard to pinpoint exactly what the experiment is aiming to measure. If it's the response to the fraction of time signaled relative to disruption, then 24 hours is an arbitrary duration, and other durations should also be considered since the absolute values do matter here. However, if the experiment is looking at response to absolute duration of the signal itself, then why not isolate that effect by measuring output at a fixed number of hours after the signaling is disrupted, say 2 hours after turning on light?

- Lines 341-357 accompanied by Figures 5F,G were very confusing! The part about persistence driving a more robust response makes complete sense, and breaking up a 6-hour stimulation into pulses of different duration and separation also makes total sense. What seems non-obvious (once again) is why the 24-hour mark is chosen to measure the end-point response, rather than some end-point with respect to the final stimulation itself. If the measurement of output were made at the end of the last pulse, or at least at a fixed duration after the last pulse ends, then there could

be an easy comparison between a single continuous stimulation versus one that's broken up by dissipative intervals. It would be possible to measure whether chaining intermittent pulses over a total duration gives a higher output than a continuous signal, even though the total signal duration is constant. But the current reported form of the experiment has the same problem pointed out before - the pulses are also subjected to different decay kinetics after the signaling period is over. So it's hard to get why something that's been allowed to dissipate over a full 18 hours should be comparable to something that's spent less time dissipating after the end of stimulation. Coupling activation and dissipation in this way really doesn't seem helpful in deriving the conclusion that is presumably sought, which is that pulsatile activation can be sustained for longer periods without saturation - very pertinent to Figure 7 and its corresponding section in the text! 24 hours seems like an arbitrary endpoint, and it's not clear that the conclusions would hold, were the end-point chosen to be 12 hours or 48 hours instead.

- Figures 5F,G themselves were hard to interpret. This is the overall understanding - that a 6-hour signal was broken down into 3, 4, 5, 6, 12, or 15 equal segments, uniformly spaced apart by different interval times. The output for each was measured at the chosen 24-hour end point, relative to the output at the same time-point for a continuous pulse. Hence, the zero-interval point always has a value of 1, being identical to a continuous signal. However, as this interval increases, the behaviour is expected to asymptote to the dotted line, because this corresponds to completely disparate individual pulses. Yet, that is not what happens in these figures! What is so puzzling is why the output actually increases back towards 1 as the refractory interval rises beyond 30 minutes, as though separating the pulses further makes them behave more and more like a continuous signal!! This is not at all intuitive, and it's not clear from the text why this should happen. It's hard to disentangle whether this is an artefact of choosing the 24-hour end-point, or something more interesting happening in this network. It would be important to address this point properly in the text.

- The 15-minute pulses are the most understandable case - their individual effects are not discernible, yet they have cumulative effects if the spacing between them is less than 10 minutes, which is also the decoherence time of the cytosolic signaling network. Also, they asymptote to the dotted line at zero with increasing inter-pulse interval, which is perfectly as expected. However, they do not actually reach that dotted line! This can be interpreted as being a result of memory being stored in the mRNA abundance, which can chain pulses over even longer periods of time. Here, it would be interesting to see if the plateau has a dependence on the actual number of 15-minute pulses chained together - which would be the case if memory were stored in mRNA. Interestingly, this plot ends at 30 minutes, which is the exact point in all the other plots where the output starts increasing towards 1 again, and also corresponds to two timescales of interest here - the half-life of mRNA, and also the minimum signal duration to see a discernible output. It would be good to see what happens to 15-minute pulses that are separated by longer intervals.

- It also seems like the effect of delays in this system - introduced because of each step needing to reach a certain threshold before the next step gets activated - may be quite important. Thus, whether pulses get chained or not may depend not only on whether the refractory interval is longer than the decoherence time, but also on whether the pulse duration itself is longer than this delay time. This could explain the discrepancy in behaviour between the 15-minute and 90/120-minute pulses observed in these plots.

- In summary, while it's interesting to see the 10-minute cytosolic decoherence time clearly delineated here in terms of its effects, it's not the only relevant timescale here - the 30-minute mRNA half-life matters too! Bringing this into the picture may be very useful in interpreting these results more clearly, and also providing deeper insights into persistence, memory, and their timescales along distal nodes in an input-output cascade. Pulsatile activation of this nature actually corresponds to strategies in digital circuits, where sawtooth patterns or chatter around a maximum can be used to maintain a high level of some output without saturating the system. This is a pretty

interesting parallel that would be great to explore, and Figure 7 already hints at its application potential.

- Another idea from digital electronic circuits that may be helpful for thinking about this section is pulse modulation. There are three such types of modulation. Pulse Amplitude Modulation is not relevant here, since all signals are considered to be saturating. However, Pulse Frequency Modulation (PFM - where pulses have the same duration but are spaced apart by different intervals) and Pulse Width Modulation (PWM - where pulses are separated by the same intervals but have different durations) are both relevant to these results, and may help organize them more clearly. The individual subplots each show a form of PFM, with their dependence on the interval between pulses. The same time-point, say 60-min, across each plot corresponds to PWM. Re-interpreting these results in terms of PFM and PWM may help disentangle the two variables, interval and duration, more clearly, and provide an easy parallel to well-understood concepts for thinking about it.

- In fact, what would also be interesting is the response to a randomly distributed set of pulses, varying in both duration and spacing, but totaling 6 hours (with a pattern looking something like a barcode on a product label) - and how different the outputs look for many such random patterns, especially relative to a continuous pulse. This would be a very succinct comparison of whether the temporal pattern of pulses matters or not, and whether having relaxation periods helps a cell sustain its activation/outputs for longer or at higher levels.

- If it so happens that this section has not been interpreted correctly here, then please be aware that it was indeed hard to read and grasp the intent of the authors, and the takeaways that are meant to be highlighted. The brevity of the writing was the biggest challenge here, demanding a lot of interpolation from the reader.

Discussion section -

3. The selection of intracellular calcium, ERK, FOS/JUN, and NFAT/CD69 for investigation has been justified by highlighting them as known nodes of interest in the TCR response signaling and activation pathway. However, a deeper discussion of their relative positions in the cascade, or their relationships to each other, would be of additional interest.

It's very interesting to think about the relationship between signal persistence timescales, "memory", and how far downstream from the receptor a certain component is. In addition, it is also nice to relate this to successive thresholds of activation down the cascade, and how they relate to persistence at each preceding step - that is, is the persistence at a given step sufficient to build up levels that sustain activation of the next step, before the signal decoheres? What are the requirements that such continuity places on the half-lives of activation and dissipation for successive steps? Why is it better to have longer persistence more distal to the receptor? It would be great to have a deeper discussion about this.

4. These timescales, especially their progression with decreasing proximity to the receptor, are particularly relevant when speculating about the role of persistence as a "filter" that can weed out noise or short spurious signals, while restricting the response to more meaningful contacts. This is closely coupled to the discussion regarding pulsatile signals and how a cell can integrate temporally-separated inputs, particularly in relation to the spacing between signals that is still able to maintain such continuity in the response. This raises the question - does a T-cell use the interval between receptor stimulation to distinguish between signal and noise? In effect, can an antigen with weaker binding strength or shorter binding times still trigger a response, if it's present at concentrations where the T-cell encounters it often and quickly enough? How can this be related to previous studies (for eg, the work by Altan-Bonnet and Germain) that look at trade-offs between

antigen binding strength and concentration? This may be out of the scope of the current work, but if included in the discussion, could potentially enrich the understanding of T-cell activation mechanisms further.

5. The usage of "opposing reactions" or "reverse steps" and references to their "efficiency" or "potency", when talking about dissipation of signals, strongly suggests that these are active processes. However, in general, this doesn't seem necessary - simple entropy could ensure that ordered associations (like signaling cascades) fall apart within the cytosolic environment, even on the rapid timescales observed in this study.

This seems significant since the manuscript discusses the idea of "memory" in the signaling network, which appears to be obviously sustained by the fact that dissipation timescales are significantly slower than activation timescales for all components discussed here - an observation that is consistent with an active process (typically faster) subsequently erased by a passive process (typically slower). In effect, the decohered state could be the stable thermal equilibrium for the cell, with active processes and the input of free energy required to assemble and sustain a coherent cascade out of the components, and everything falling apart quickly once that active input is removed.

If these are indeed active reactions, it may be helpful to specify that explicitly, and provide some examples of these processes in the discussion. If the decoherence is more likely to be from thermodynamically driven passive processes, then it may be better to change the wording to "thermal dissipation" of components in the network, and avoid references to process "efficiency".

A general comment -

6. Although the manuscript is very well-written for the most part, some of the sentences are long and complex. They would benefit from being broken down into simpler shorter pieces, each capturing only a single concept.

Reviewer #2:

The work by Harris, Fuyal, and James uses optogenetics to investigate temporal signal integration by the T cell signalling network using an elegant experimental system. Their key finding is that there is virtually no signal persistence at the level of signalling (30 s for Ca and 3 min for ERK) and only ~15 minutes at the level of mRNA. They also suggest that short signal pulses produce stronger signals compared to constant signals as a result of feedbacks that engage under constant signals.

The work includes clear figures and text, and is logically developed making it very easy to read. The conclusions are novel and very exciting and for me, it is the first study that takes advantage of the key novelty of optogenetics, namely to investigate signal reversibility in detail.

I have some minor suggestions that I hope will improve the work even further

Less minor suggestions:

1. The work is motivated based on observations that T cells integrate signals across APCs in the introduction and then argues that their results are inconsistent with these notions. To clearly

demonstrate this, can the authors explicitly state a timescale in the introduction? For example, in previous studies, did signal integration across APCs require the signal to persist for hours? If they make this clear, it will then be very easy to convince the reader that the 15 minutes of mRNA persistence can't do it.

2. There have been previous reports that showed that constant receptor signalling is required for cytokine secretion (Huppa et al (2001) Nature Immunology) and that this depends on co-stimulation (Trendel et al (2020) biorxiv). This was always puzzling because it was thought that there is some signal persistence, and the current study nicely solves this puzzle by showing that in fact, signal persistence is very weak. It is worth noting that the mRNA persistence for 15-30 minutes is possibly discrepant with these studies. I wonder if the authors can include a comment on whether cytokine secretion may require continuous receptor signalling even if mRNA persists?

3. In the data in Figure 7C the authors argue that CD137 expression stops increasing between 4-12 hours is a result of feedback inhibition but I didn't follow why this then disappears at 12 hours so that CD137 can then continue to increase. I wondered if CD137 could operate like CD69 with pre-formed CD137 reaching the surface early with the second rise associated with transcriptional activity? If receptor signalling is needed for both to reach the surface, could this also explain the mechanism of the pulsatile signals?

More Minor comments:

1. Page 3: "It is therefore unlikely that T cells directly integrate TCR signals between multiple APC interaction, but rather accumulate the transcriptional output of gene expression from each stimulation"

This is semantic but how does the accumulation of transcriptional output not integrating TCR signals? I think it can be argued that it is a mechanism. I agree that it is surprising, given that this integration is generally believed at the level of signalling.

2. Line 175: I suggest changing 'convinced ourselves' to 'shown' since you also have convinced the reader by this point!

3. Line 195: The statement that the reduction in calcium is consistent with five steps in the main text could be more detailed. What assumptions underlie how they arrive at these 5 steps and why is it important? Can the author analyse their ERK and mRNA data similarly?

4. Line 316-318: The authors state that they do not see saturation but intensity of both GFP and CD69 appear to saturate after about 8 hours in Fig 5A/C? Obviously not in B/D which only show shorter signal durations

5. Fig 7B: Why do not all the cells up regulate CD137 in the dark? Is this a result of the transduction efficiency being below 100%?

6. Fig 7F: It would be easier to follow with a legend on the panels

7. There has been suggestion that T cell signalling can exhibit hysteresis in a high profile paper (Das et al (2009) Cell) using the same Jurkat system that the authors have used. This is not central to their paper but it occurs to me that their experimental system, and data they already acquired, can confirm or refute this idea based on experiments similar to those in Fig S4B/C. Fig S4 currently shows the path when removing the signal and if a similar graph is made for providing the signal,

then it will be very clear if hysteresis is present. This analysis would resolve whether the 'digital' nature of T cell signalling is bistable or ultra-sensitive. I want to emphasise that this is a comment and not necessarily something that the authors would want to include in the present manuscript.

Reviewer #3:

The authors developed a system to control signaling in an engineered immune receptor with high temporal sensitivity to study the dynamics of signaling pathways. To do so they used a small molecule inducible on-type switch system on a CAR related receptor design to mediate cell-cell interactions on the extracellular side and used the LOV based optogenetic system for light induced dissociation of the intracellular signaling domain. The on switch serves to synchronize the start of signaling and the off switch serves to shut off signal activation to investigate dynamics of downstream effects. The underlying biological questions, in particular precise measurements of signal persistence, are important and relevant and the experimental design appears to be suitable to answer some of these questions. The methods section is excellent and especially the tabular listing of reagents is very clear. Unfortunately, at the current state of the manuscript the extensive data normalizations make the interpretation of the results not verifiable and a lot of underlying information is lost. Many parts of the paper require additional explanations and many claims need to be more robustly backed up by references and data. I think the manuscript contains valuable information that is worth the effort of extensive clarifications.

In detail:

Major point for almost all of the figures: Please also provide non normalized values and define 0 and 1. All different illumination regimes also require a control for possible light induced effects.

Line 17-19: claim is not sufficiently backed up by data of the manuscript.

29-30: Defining output of receptor signaling as 'normally manifested [...] as changes in gene expression' is too narrow. Gene expression is only a fraction of responses, especially in the context of T-cells (e.g. vesicle secretion). Please provide a more comprehensive and referenced definition.

85: "Photoswitchable CAR-T Cell Function In Vitro and In Vivo via a Cleavable Mediator" that likely appeared at a similar time as your submission is another related paper that should be discussed here.

86: Clarify 'TCR-based receptor'. In the rest of the document, you refer to it as CAR. Please explain the differences to a CAR and why you chose this particular design.

91-93: Please give some explanation why a short half-life of signaling events speaks against direct TCR signal integration from multiple receptors and co-receptors. Also, please discuss if your system is sufficient to generate data to answer this question.

108-110: Please discuss in more detail in which way this receptor more closely replicates TCR complex function than other CAR designs and what are limitations of it.

114-116: How does this 'most closely replicate the response to' TCR dissociation? Please give some reasons to choose intracellular or extracellular disruption for this question.

125-127: Here you convincingly argue for studying single signaling pathways in isolation. However, please reconcile this statement with your ambition of studying multiple APC - T cell interactions (compare 92) and clarify these statements accordingly.

153-156: This section needs more explanations. Why does proximal signaling remain uninitiated? Does other signaling occur?

167-170: This is a key control! It is unclear if this control was included for other experiments as well. E.g. for the same setup as 1 D and E and for other signaling pathways, this is of major importance to

exclude light induced effects. In some figures different LOV2 domain mutants were used but not clearly labeled. It would help clarity to always include the name of the LOV2 mutation in the respective figure. Please provide also non-normalized measurements for comparison and explain how you define 0 and 1?

229: Figure 3B, 3E- did you perform negative controls with the LOV2 mutant for this experiment?

230-230: Please discuss this statement also in the context of non-normalized values. For the whole figure 3, from the figure alone it's difficult to understand what was done. Time of induction with the dimerizer and time of illumination should be more clearly labeled.

264-266: Due to all the normalizations in the data analysis in this figure I cannot make any comments on the relevance or accuracy of any of the conclusions.

300: reference missing

301: reference missing

316: Figure 5 All of the normalized graphs should also be compared to non normalized data and discussed in that context.

352: Figure 5: x on the light intervals is not explained. E.g. 'light induced OFF' might be clearer. Also the relative lengths of on/off should be indicated. The blue is getting lighter to the right, giving the impression that light intensity was reduced, which is probably not intended, correct? The X label is unclear, shouldn't it be minutes instead of interval (min)? From the figure alone I cannot make out what it's supposed to show. The normalizations again make interpretations very difficult.

375: Please explain why this is expected and provide a reference.

382-386: Possibly except for IL2 all of the graphs in A show downregulation independent on illumination. Especially for FOS. Please explain your results in the context of this. Please also discuss other possibilities than mRNA half-life that could play a role.

399: Reference missing

406: Please define 'rate of activation' and explain how your data supports negative feedback and why signaling pulses may be beneficial.

411-413: Doesn't this mean that continuous induction was better than pulsed after all? Please clarify. Different populations should be labeled in the figure, best with color codes. Extracting that information from the legends alone is difficult.

419-421: What does this mean for a real TCRs or a CARs? What are limitations in these interpretations?

422: Is this statement pathway specific?

429-431: I don't see this claim justified by the data in the current form.

432-433: Please discuss how your measurements justify to make claims regarding multiple APC interactions and limitations of them.

455-457: This interpretation strikes me as somewhat simplistic. Could you discuss other factors that contribute to persistent cellular changes caused by signaling?

482-484: Why do you argue that the OportoCAR is more equivalent to the TCR than other sorts of CARs and in what regards does it fall short?

592: The normalization is very unclear here, please show this plot without normalization for comparison. By eye there seems to be no very marked difference in FOS in the measured time, unlike what this plot shows and also unlike what panel E and figure 6 shows. Please discuss these discrepancies.

605: What is the baseline here? The normalization should be explained better. Please include the non-normalized values for comparison and explain how you define 0 GFP in this plot.

615: Please indicate in the figure the intervals for both light on and light off. From figure and legend it is unclear what the dotted line describes. The first pulse only and then light? At what time point is it shown?

627: what is this 'defined period'? 3 hours?

641: Please define 'control'

644-645: Please define what you mean by '24-hour range of signal pulses.' And compare to non normalized data.

885-886: I don't understand this sentence. Please clarify.

913: Please also plot the total RNA levels relative to GAPDH and PGK1 as a comparison to your normalized data.

927-928: This can be very misleading. Please show the graphs with the cRQ in the main figure. As of now, I cannot comment on any relevance or interpretation of the data, as normalization removes every scale of the effects. Please justify your choice of calculating mRNA half-life based on this normalized data.

982-985: Please explain in more detail what you did here and how you interpret the results

989: Could you explain the purpose and your interpretation of this plot? Can you see the bands corresponding to protein of the upper panel or are they too faint?

1001: Could you please show data for baseline expression compared to WT cells?

1020: This format gives much more relevant information. These plots really do look very different. Please discuss the uses of the normalized data for the analysis and justify how and when normalization is beneficial for data analysis and when it isn't.

Reviewer #1:

Summary of the paper –

This well-written manuscript presents an interesting series of results dissecting receptor-proximal signal persistence, timescales, and additivity, in a system engineered to provide dual chemical and optogenetic control of T-cell antigen receptor signaling.

The contributions are well-laid out, with a clear logical sequence as follows -

1. A description of the engineered OptoCAR where light-controlled separable domains are inserted between the membrane-bound receptor and its intracellular signaling domains, accompanied by a validation of its expected behaviour in non-immune cells and characterization of light-reversible association-dissociation kinetics.

2. Investigation of successively distal constituents of the signaling network downstream of the receptor, showing the elongation of signal dissipation timescales further down the signaling cascade

(i) Intracellular calcium fluxes, which dissipate rapidly within seconds of receptor deactivation

(ii) Receptor-proximal cytosolic signaling by ERK, followed by nuclear localization of FOS/JUN, which preserve memory on the short timescale of a few minutes

(iii) De novo gene expression of NFAT-driven GFP and CD69, which translates these signals into outputs, with a persistence determined by mRNA half-life scales of up to 30 minutes

(iv) Further comparison with mRNA expression levels of other response genes, showing effects of temporal coupling between signaling and transcription

3. A comparison between continuous and pulsatile signaling at distal stages having longer persistence, showing the downstream effects of interaction between pulses.

4. Application of such pulsatile signaling to a clinically-relevant CAR, where the spacing between signals sustains activity over time by allowing relaxation of the signaling network, thus avoiding saturation and negative feedback loops.

Some of the major strengths of the paper -

1. Although the paper covered diverse technical and biological aspects - such as the biology of TCR receptor signaling itself and its downstream components, the design of the optogenetic system and associated experimental platforms for tuned illumination, and a practical application in the context of a clinically relevant CAR - the background for each aspect was laid out well enough for any reader not expert in a particular area to still be able to place results in context and appreciate the work as a whole.

2. The requisite controls for all experiments have been considered thoroughly, and were comprehensively and convincingly presented in the supplements to bolster all the major claims.

3. The paper appears technically very sound, with detailed discussion of the observations, clear understandable figures, and well-supported conclusions.

We thank the author very much for their time in reviewing the manuscript, and for helping us to improve our work. We hope we have addressed all their points below to their satisfaction.

Questions, comments, and suggestions for the manuscript -

Overall, the technical aspects and results are mostly well-presented. The discussion section may have some more potential for depth, since conclusions in this work can be considered in some more interesting directions.

Results section -

(1) It might be easier for a reader to first see Figure 6 and the section titled "Gene transcription is rapidly abolished after disruption of receptor signaling", prior to Figure 5 and the section "Short-term signal persistence can be directly observed in gene expression output".

- Figure 6 and the mRNA expression time-courses serve as a nice extension to the work presented on calcium signaling, FOS, and ERK, with transcription being the next logical step in the cascade, and the half life being extended from ~10min to ~30min. This also showcases mRNA production as a step that extends the "memory" in the system for a while longer, facilitating the chaining of signals separated by longer period of time, which have already decohered at earlier nodes in the network. With this background in place first, it becomes easier to grasp why integration is possible, even when the pulses are individually undetectable.

- Figure 5 extends that by one step, to the outputs of transcription and translation - that is, observable protein levels of GFP and CD69, which may be considered the final output of this cascade. Having the mRNA information in place before looking at protein expression is just easier to follow, and also consistent with the layout of this manuscript, presenting successively distal processes from the receptor.

- Figure 5 is also a better segue to Figure 7, connecting the concepts through pulsatile signaling.

This is a good suggestion and we have now rearranged the flow of the manuscript as suggested.

(2) The section on "Short-term signal persistence can be directly observed in gene expression output" could be much more clearly presented and explained to the reader. There are some aspects that are really not clear, or possibly very interesting but insufficiently addressed.

We have now rewritten parts of this section and included some additional schematics based on the reviewer's suggestions below. We hope this part of the manuscript is now clearer.

- Subplot 5E would ideally come first in this figure, since it shows the expression kinetics over time, which is the baseline for everything else discussed here - saturation, pulses, etc.

This panel has now been moved to the EV figure and referenced before the pulse duration to improve the flow of the manuscript.

- For Figures 5A-D and Lines 308-311, it appears that the total signal duration ranges up to 24 hours total, but only end-point expression is measured, uniformly at the end of 24 hours in all cases? This would be helpful to specify very explicitly in the text, with a clear explanation of why this design was chosen (i.e. why 24-hours).

We have now improved this section to make it clear why a single endpoint is used for these experiments. 24 hours is a 'standard' period for measuring the output from T cell activation. This point is discussed further below.

- In Figures 5A-E, and Lines 315-318, why is there a broad claim that the system is not saturating? This seems clearly contradicted by Fig 5E, where the expression curve is obviously reaching saturation for signal durations approaching 24 hours, and Figs 5A,C where the population appears to show saturation in the number of responders (and their intensity) beyond about 10 hours. Also, Figs

5B,D which do not show saturation are on a different X-axis scale of only 6 hours - why is this also not extended to 24 hours? It's true that there's no saturation within this 6-hour regime (which is probably very relevant for the next steps in Figs 5F,G, and maybe why this is highlighted explicitly in the text), but there's clearly saturation over a 24-hour signaling period! This should be properly disentangled, to be consistent throughout and less confusing.

We thank the reviewer for pointing this out, which was an error in redrafting the manuscript when it referred solely to the 6-hour period. This section has now been revised and this statement corrected. The 6-hour plot was simply a subset of the full 24-hour plot presented in the supplementary figure. To avoid this confusion, only the full 24-hour dataset is now presented but the lack of saturation over 6 hours is explicitly stated for exactly the reason the reviewer mentions.

- In Lines 321/339, please quantify what is meant by a detectable pulse - is it a 5% increase in expression level above the baseline, or just the first increase of any magnitude? How is signal distinguished from noise at this low level, and how does the 30-minute level compare with the variations observed in these measurements?

This is a good point; we now explicitly state this is a significant increase in output above background at $p < 0.05$, to be more quantitative.

- In Figures 5A-D, when the end-point was measured, it was hard to tell if this is just at the end of the dark signaling period, or the end-point was after a full 24 hours? If it's the former, then the end-point is related to the signal duration. But if it's the latter, then the outcome is a consequence of both activation in the dark period and dissipation during the light period, both of which have varying durations that cannot be decoupled. In this case, it's hard to pinpoint exactly what the experiment is aiming to measure. If it's the response to the fraction of time signaled relative to disruption, then 24 hours is an arbitrary duration, and other durations should also be considered since the absolute values do matter here. However, if the experiment is looking at response to absolute duration of the signal itself, then why not isolate that effect by measuring output at a fixed number of hours after the signaling is disrupted, say 2 hours after turning on light?

We spent a substantial amount of time trying to think of the best way to do these experiments in a controlled manner. For figures 5A-D, the expression is measured at the end of 24 hours, with the intervening time from the end of the dark state under constant illumination. If we measured straight after the signal pulse, this would simply be a time course. Our rationale was to provide the expression profile for a single pulse that would be directly comparable to the subsequent PWM experiments. This does have the advantage of ensuring that all potential output is detected, assuming a delay between mRNA production and GFP maturation or CD69 trafficking to cell surface.

The reviewer has a very good point that we did not explicitly show in the paper the dissipation of outputs in the refractory light period. To address this, we have now included two new pieces of supplementary data to show that the decay of CD69 and especially GFP outputs are very low, so that the measured values at the end of 24 hours are a good reflection of what the maximal response was from the single pulse of signalling. We hope this goes some way to alleviate the reviewer's concerns.

We did consider the approach of measuring a fixed period after the varying single pulse length, but this would make the data less comparable to the PWM datasets, for which this approach would be practically impossible. We needed a way to take samples at one timepoint and this was the best means.

- Lines 341-357 accompanied by Figures 5F,G were very confusing! The part about persistence driving a more robust response makes complete sense, and breaking up a 6-hour stimulation into pulses of different duration and separation also makes total sense. What seems non-obvious (once again) is why the 24-hour mark is chosen to measure the end-point response, rather than some end-point with respect to the final stimulation itself. If the measurement of output were made at the end of the last pulse, or at least at a fixed duration after the last pulse ends, then there could be an easy comparison between a single continuous stimulation versus one that's broken up by dissipative intervals. It would be possible to measure whether chaining intermittent pulses over a total duration gives a higher output than a continuous signal, even though the total signal duration is constant. But the current reported form of the experiment has the same problem pointed out before - the pulses are also subjected to different decay kinetics after the signaling period is over. So it's hard to get why something that's been allowed to dissipate over a full 18 hours should be comparable to something that's spent less time dissipating after the end of stimulation. Coupling activation and dissipation in this way really doesn't seem helpful in deriving the conclusion that is presumably sought, which is that pulsatile activation can be sustained for longer periods without saturation - very pertinent to Figure 7 and its corresponding section in the text! 24 hours seems like an arbitrary endpoint, and it's not clear that the conclusions would hold, were the end-point chosen to be 12 hours or 48 hours instead.

Some of these important points have been addressed in the response above. We have modified the text significantly to try and make these points clear and our reasons for choosing the approach we have. The benefit of fixing the experiment to 24 hours is that it also fixes the total illumination time of the samples to 18 hours. We have added some further supplementary data to show that there is a weak phototoxic effect on OptoCAR T cells, so controlling for total light exposure removes this as a confounding effect in the datasets. We hope that our new supplementary data on the low dissipation of outputs helps to alleviate these concerns.

However, to provide further confidence to the reviewer that our results at 24 hours stand for other endpoints, we have now repeated the entire PWM datasets using a 12-hour endpoint instead, included in the EV figure. The two datasets are strikingly similar and so we believe this is good evidence our approach is a valid one.

- Figures 5F,G themselves were hard to interpret. This is the overall understanding - that a 6-hour signal was broken down into 3, 4, 5, 6, 12, or 15 equal segments, uniformly spaced apart by different interval times. The output for each was measured at the chosen 24-hour end point, relative to the output at the same time-point for a continuous pulse. Hence, the zero-interval point always has a value of 1, being identical to a continuous signal. However, as this interval increases, the behaviour is expected to asymptote to the dotted line, because this corresponds to completely disparate individual pulses. Yet, that is not what happens in these figures! What is so puzzling is why the output actually increases back towards 1 as the refractory interval rises beyond 30 minutes, as though separating the pulses further makes them behave more and more like a continuous signal!! This is not at all intuitive, and it's not clear from the text why this should happen. It's hard to disentangle whether this is an artefact of choosing the 24-hour end-point, or something more interesting happening in this network. It would be important to address this point properly in the text.

We apologise for the confusion that the dotted line has made in these figures, which have now removed. This line indicated the response for a single pulse (not to total of disparate individual pulses) but we see that this was very unhelpful to the comprehension of the figure. We now also explicitly mention the 'smiling' of the datasets, which is primarily observed for the CD69 datasets. It

is not an artefact of the 24-hour endpoint, as it is also observed in the new 12-hour endpoint dataset too. It is very likely that the more observable dissipation/decay of CD69 means that pulses that finish nearer to the experimental endpoint (i.e. points on RHS of plots) have dissipated less and so have a higher measured output than those in the 'middle' of the plots. We did some simple modelling to show that this argument made sense but didn't think it was appropriate to add into the manuscript. This effect does not alter our conclusions about the fast decay of signal coherence, so we hope that the reviewer is happy with our interpretation and text changes.

- The 15-minute pulses are the most understandable case - their individual effects are not discernible, yet they have cumulative effects if the spacing between them is less than 10 minutes, which is also the decoherence time of the cytosolic signaling network. Also, they asymptote to the dotted line at zero with increasing inter-pulse interval, which is perfectly as expected. However, they do not actually reach that dotted line! This can be interpreted as being a result of memory being stored in the mRNA abundance, which can chain pulses over even longer periods of time. Here, it would be interesting to see if the plateau has a dependence on the actual number of 15-minute pulses chained together - which would be the case if memory were stored in mRNA. Interestingly, this plot ends at 30 minutes, which is the exact point in all the other plots where the output starts increasing towards 1 again, and also corresponds to two timescales of interest here - the half-life of mRNA, and also the minimum signal duration to see a discernible output. It would be good to see what happens to 15-minute pulses that are separated by longer intervals.

Some of these points are addressed above. We actually collected data beyond 30 minutes for the 15 minute dataset, up to the maximum 45 minute interval (and eqv for other pulse durations). We did not include to more clearly emphasize the initial decay. However, given the reviewer's comments, we have now included the full datasets. In combination with the points above, it should now be clearer that the asymptote for these datasets is the sum of independent pulses.

- It also seems like the effect of delays in this system - introduced because of each step needing to reach a certain threshold before the next step gets activated - may be quite important. Thus, whether pulses get chained or not may depend not only on whether the refractory interval is longer than the decoherence time, but also on whether the pulse duration itself is longer than this delay time. This could explain the discrepancy in behaviour between the 15-minute and 90/120-minute pulses observed in these plots.

We take this point on board and try to include it in the discussion.

- In summary, while it's interesting to see the 10-minute cytosolic decoherence time clearly delineated here in terms of its effects, it's not the only relevant timescale here - the 30-minute mRNA half-life matters too! Bringing this into the picture may be very useful in interpreting these results more clearly, and also providing deeper insights into persistence, memory, and their timescales along distal nodes in an input-output cascade. Pulsatile activation of this nature actually corresponds to strategies in digital circuits, where sawtooth patterns or chatter around a maximum can be used to maintain a high level of some output without saturating the system. This is a pretty interesting parallel that would be great to explore, and Figure 7 already hints at its application potential.

We take this point on board and try to include it in the discussion.

- Another idea from digital electronic circuits that may be helpful for thinking about this section is pulse modulation. There are three such types of modulation. Pulse Amplitude Modulation is not relevant here, since all signals are considered to be saturating. However, Pulse Frequency

Modulation (PFM - where pulses have the same duration but are spaced apart by different intervals) and Pulse Width Modulation (PWM - where pulses are separated by the same intervals but have different durations) are both relevant to these results, and may help organize them more clearly. The individual subplots each show a form of PFM, with their dependence on the interval between pulses. The same time-point, say 60-min, across each plot corresponds to PWM. Re-interpreting these results in terms of PFM and PWM may help disentangle the two variables, interval and duration, more clearly, and provide an easy parallel to well-understood concepts for thinking about it.

We have now explicitly used the term PFM for our datasets in the manuscript and have also provided a schematic to illustrate PFM; we thank the reviewer for prompting us to formalise this in the main text. The schematic should also significantly aid readers understand the context of the experiments. We did try to 'reslice' the data in the context of PWM but there are insufficient points of equivalent frequency to make this useful.

- In fact, what would also be interesting is the response to a randomly distributed set of pulses, varying in both duration and spacing, but totaling 6 hours (with a pattern looking something like a barcode on a product label) - and how different the outputs look for many such random patterns, especially relative to a continuous pulse. This would be a very succinct comparison of whether the temporal pattern of pulses matters or not, and whether having relaxation periods helps a cell sustain its activation/outputs for longer or at higher levels.

We put some thought into performing this experiment but realised there would be no natural independent variable with which to plot our data against. So, we would assume to get some distribution of output varying about a mean value. We were unclear what further insight this would give us beyond what we have already presented. We instead focused our efforts on performing the PFM experiment at a shorter endpoint (discussed above) as this seemed more pertinent.

- If it so happens that this section has not been interpreted correctly here, then please be aware that it was indeed hard to read and grasp the intent of the authors, and the takeaways that are meant to be highlighted. The brevity of the writing was the biggest challenge here, demanding a lot of interpolation from the reader.

We have tried to improve the readability of this section based on the reviewer's very helpful comments, and we hope it is now significantly improved.

Discussion section -

3. The selection of intracellular calcium, ERK, FOS/JUN, and NFAT/CD69 for investigation has been justified by highlighting them as known nodes of interest in the TCR response signaling and activation pathway. However, a deeper discussion of their relative positions in the cascade, or their relationships to each other, would be of additional interest.

We have used the schematic in the first figure to highlight more clearly their relative positions in the cascade and discussed how different phosphorylation targets (Tyr vs S/T) might influence the dissipation rates of the signals.

It's very interesting to think about the relationship between signal persistence timescales, "memory", and how far downstream from the receptor a certain component is. In addition, it is also nice to relate this to successive thresholds of activation down the cascade, and how they relate to persistence at each preceding step - that is, is the persistence at a given step sufficient to build up levels that sustain activation of the next step, before the signal decoheres? What are the

requirements that such continuity places on the half-lives of activation and dissipation for successive steps? Why is it better to have longer persistence more distal to the receptor? It would be great to have a deeper discussion about this.

We have included some discussion on this point.

4. These timescales, especially their progression with decreasing proximity to the receptor, are particularly relevant when speculating about the role of persistence as a "filter" that can weed out noise or short spurious signals, while restricting the response to more meaningful contacts. This is closely coupled to the discussion regarding pulsatile signals and how a cell can integrate temporally-separated inputs, particularly in relation to the spacing between signals that is still able to maintain such continuity in the response. This raises the question - does a T-cell use the interval between receptor stimulation to distinguish between signal and noise? In effect, can an antigen with weaker binding strength or shorter binding times still trigger a response, if it's present at concentrations where the T-cell encounters it often and quickly enough? How can this be related to previous studies (for eg, the work by Altan-Bonnet and Germain) that look at trade-offs between antigen binding strength and concentration? This may be out of the scope of the current work, but if included in the discussion, could potentially enrich the understanding of T-cell activation mechanisms further.

We have included some discussion on this point.

5. The usage of "opposing reactions" or "reverse steps" and references to their "efficiency" or "potency", when talking about dissipation of signals, strongly suggests that these are active processes. However, in general, this doesn't seem necessary - simple entropy could ensure that ordered associations (like signaling cascades) fall apart within the cytosolic environment, even on the rapid timescales observed in this study.

This seems significant since the manuscript discusses the idea of "memory" in the signaling network, which appears to be obviously sustained by the fact that dissipation timescales are significantly slower than activation timescales for all components discussed here - an observation that is consistent with an active process (typically faster) subsequently erased by a passive process (typically slower). In effect, the decohered state could be the stable thermal equilibrium for the cell, with active processes and the input of free energy required to assemble and sustain a coherent cascade out of the components, and everything falling apart quickly once that active input is removed.

If these are indeed active reactions, it may be helpful to specify that explicitly, and provide some examples of these processes in the discussion. If the decoherence is more likely to be from thermodynamically driven passive processes, then it may be better to change the wording to "thermal dissipation" of components in the network and avoid references to process "efficiency".

This is a very interesting perspective. We have tried to be clearer now that the activation steps are generally irreversible at the kinetic level (predominantly phosphorylation) but the opposing dephosphorylation reactions must be active and efficient. It is unlikely that reversion to thermal eqm is the driving force behind the reversal of the signal. In support of this, simply inhibiting phosphatases in resting T cells is sufficient to drive activation in the absence of ligand binding, implying continuous enzymatic activity is required to maintain the quiescent state. We have tried to touch on these points in the discussion.

A general comment -

6. Although the manuscript is very well-written for the most part, some of the sentences are long and complex. They would benefit from being broken down into simpler shorter pieces, each capturing only a single concept.

We have refreshed the manuscript to improve its readability.

Reviewer #2:

The work by Harris, Fuyal, and James uses optogenetics to investigate temporal signal integration by the T cell signalling network using an elegant experimental system. Their key finding is that there is virtually no signal persistence at the level of signalling (30 s for Ca and 3 min for ERK) and only ~15 minutes at the level of mRNA. They also suggest that short signal pulses produce stronger signals compared to constant signals as a result of feedbacks that engage under constant signals.

The work includes clear figures and text, and is logically developed making it very easy to read. The conclusions are novel and very exciting and for me, it is the first study that takes advantage of the key novelty of optogenetics, namely to investigate signal reversibility in detail.

I have some minor suggests that I hope will improve the work even further

We thank the author very much for their time in reviewing the manuscript, and for helping us to improve our work. We hope we have addressed all their points below to their satisfaction.

Less minor suggestions:

1. The work is motivated based on observations that T cells integrate signals across APCs in the introduction and then argues that their results are inconsistent with these notions. To clearly demonstrate this, can the authors explicitly state a timescale in the introduction? For example, in previous studies, did signal integration across APCs require the signal to persist for hours? If they make this clear, it will then be very easy to convince the reader that the 15 minutes of mRNA persistence can't do it.

Thank you for this suggested inclusion, which as the reviewer states, will help the reader put the potential for serial APC encounters on a more quantitative footing. This point has now been made in the introduction and discussion sections.

2. There have been previous reports that showed that constant receptor signalling is required for cytokine secretion (Huppa et al (2001) Nature Immunology) and that this depends on co-stimulation (Trendel et al (2020) biorxiv). This was always puzzling because it was thought that there is some signal persistence, and the current study nicely solves this puzzle by showing that in fact, signal persistence is very weak. It is worth noting that the mRNA persistence for 15-30 minutes is possibly discrepant with these studies. I wonder if the authors can include a comment on whether cytokine secretion may require continuous receptor signalling even if mRNA persists?

We have added a comment to the discussion section to mention these studies in the context of sustained signaling and co-stimulation. Because we have not measured cytokine protein production specifically in the text we felt we could not comment either way about whether cytokines might require continuous signaling. Neither of the suggested referenced appeared to measure mRNA levels so there was no explicit discrepancy that we felt needed to be mentioned. We hope this is acceptable to the reviewer.

3. In the data in Figure 7C the authors argue that CD137 expression stops increasing between 4-12 hours is a result of feedback inhibition but I didn't follow why this then disappears at 12 hours so that CD137 can then continue to increase. I wondered if CD137 could operate like CD69 with pre-formed CD137 reaching the surface early with the second rise associated with transcriptional activity? If receptor signalling is needed for both to reach the surface, could this also explain the mechanism of the pulsatile signals?

We thank the reviewer for raising this point. To address whether this is the case, we have performed RT-qPCR on primary T cells to measure *CD137* mRNA levels before and after activation, in comparison with HEK293T and Raji cells. We find no evidence for a pre-existing pool of *CD137* mRNA prior to stimulation, inferring that no *CD137* protein is present within the T cells. We hope to follow up the mechanism for enhanced expression with pulsatile signalling in the near future. The text has been changed to include this point.

More Minor comments:

1. Page 3: "It is therefore unlikely that T cells directly integrate TCR signals between multiple APC interaction, but rather accumulate the transcriptional output of gene expression from each stimulation"

This is semantic but how does the accumulation of transcriptional output not integrating TCR signals? I think it can be argued that it is a mechanism. I agree that it is surprising, given that this integration is generally believed at the level of signalling.

This point was also raised by another reviewer. We have now changed this sentence to state that our results quantify the temporal window over which signal integration could occur, including from mRNA production.

2. Line 175: I suggest changing 'convinced ourselves' to 'shown' since you also have convinced the reader by this point!

Text changed accordingly

3. Line 195: The statement that the reduction in calcium is consistent with five steps in the main text could be more detailed. What assumptions underlie how they arrive at these 5 steps and why is it important? Can the author analyse their ERK and mRNA data similarly?

This part of the manuscript has been removed as it did not add significantly to the understanding of the results. To the reviewer's second point, the ERK and mRNA were not of sufficient temporal resolution to provide a confident fit to a specific number of steps, hence was not presented.

4. Line 316-318: The authors state that they do not see saturation but intensity of both GFP and CD69 appear to saturate after about 8 hours in Fig 5A/C? Obviously not in B/D which only show shorter signal durations

We thank the reviewer for pointing this out, which was an error in redrafting the manuscript when it referred solely to the 6 hour period. This section has now been revised and this statement corrected.

5. Fig 7B: Why do not all the cells up regulate *CD137* in the dark? Is this a result of the transduction efficiency being below 100%?

Cells are gated on OptoCAR expression so should be all transduced. All cells have upregulated *CD137* to some extent but the expression was more variable for these controls than observed in the actual experiment (compare Figure 7F plots). This may be due to lower overall OptoCAR expression in early trials. To remove this source of confusion, we have repeated this experiment and have now have control data that matches more closely to the remaining datasets in the figure.

6. Fig 7F: It would be easier to follow with a legend on the panels

This has now been added.

7. There has been suggestion that T cell signalling can exhibit hysteresis in a high profile paper (Das et al (2009) Cell) using the same Jurkat system that the authors have used. This is not central to their paper but it occurs to me that their experimental system, and data they already acquired, can confirm or refute this idea based on experiments similar to those in Fig S4B/C. Fig S4 currently shows the path when removing the signal and if a similar graph is made for providing the signal, then it will be very clear if hysteresis is present. This analysis would resolve whether the 'digital' nature of T cell signalling is bistable or ultra-sensitive. I want to emphasise that this is a comment and not necessarily something that the authors would want to include in the present manuscript.

This is indeed a very interesting suggestion and one we hope to follow up in the near future. As the reviewer states, there is limited scope to delve into this point too deeply in the current manuscript. We have included the signal titration data in Fig S4B/C as a separate section, to make it clearer that the datasets derive from continuous light at varying intensity, so it mimics a constant, graded input to the network. Therefore, we are not per se “removing the signal” but simply titrating it and measuring a fixed timepoint. These plots have now been reversed and more clearly labelled to address this point.

Reviewer #3:

The authors developed a system to control signaling in an engineered immune receptor with high temporal sensitivity to study the dynamics of signaling pathways. To do so they used a small molecule inducible on-type switch system on a CAR related receptor design to mediate cell-cell interactions on the extracellular side and used the LOV based optogenetic system for light induced dissociation of the intracellular signaling domain. The on switch serves to synchronize the start of signaling and the off switch serves to shut off signal activation to investigate dynamics of downstream effects. The underlying biological questions, in particular precise measurements of signal persistence, are important and relevant and the experimental design appears to be suitable to answer some of these questions. The methods section is excellent and especially the tabular listing of reagents is very clear. Unfortunately, at the current state of the manuscript the extensive data normalizations make the interpretation of the results not verifiable and a lot of underlying information is lost. Many parts of the paper require additional explanations and many claims need to be more robustly backed up by references and data. I think the manuscript contains valuable information that is worth the effort of extensive clarifications.

We thank the author very much for their time in reviewing the manuscript, and for helping us to improve our work. We hope we have addressed all their points below to their satisfaction. The reviewer's principal issue was with a perceived over-normalisation of our datasets. This is likely due to our over-use of the word 'normalisation'; for essentially all datasets this simply refers to the removal of a background value and scaling the whole datasets to a single reference point. We believe this is a standard way to present our data; to make this more clear, we now use the phrase 'Relative' to show the data is simply scaled to a given point. We feel this is justified because for almost all the assays, the absolute values provide no additional meaning to the datasets. However, by scaling to a particular value that is relevant to the dataset, this provides the reader a concise way to gauge the impact of light treatment, say. However, for the FOS datasets the reviewer's point is well made and we have now added the unnormalized data to the main figure to make it clear the effect is readily observable. We have now included the raw data in separate figures for all assays to help the reader.

In detail:

Major point for almost all of the figures: Please also provide non normalized values and define 0 and 1. All different illumination regimes also require a control for possible light induced effects.

We have now made clear in the figure legends how the 0 and 1 bounds are defined. As noted above, '0' is invariably just the background value but the '1' has been provided when not clear from the figure itself. We have also used the C450G and I539E OptoCAR variants in all assays now, to show the effect of light toxicity without the confounding effect of signal disruption. These datasets are provided in the Extended View Figures.

Line 17-19: claim is not sufficiently backed up by data of the manuscript.

This sentence has been modified to highlight that T cells are most likely to accumulate outputs rather than intracellular signals, based on our datasets.

29-30: Defining output of receptor signaling as 'normally manifested [...] as changes in gene expression' is too narrow. Gene expression is only a fraction of responses, especially in the context of T-cells (e.g. vesicle secretion). Please provide a more comprehensive and referenced definition.

This point is well taken. We have now broadened this sentence and added representative references.

85: "Photoswitchable CAR-T Cell Function In Vitro and In Vivo via a Cleavable Mediator" that likely appeared at a similar time as your submission is another related paper that should be discussed here.

Agreed, reference added.

86: Clarify 'TCR-based receptor'. In the rest of the document, you refer to it as CAR. Please explain the differences to a CAR and why you chose this particular design.

The phrase 'TCR-based receptor' was meant to help the reader understand the similarities between TCR and CARs; clearly not very effectively. We have now amended this sentence to make it clear our OptoCAR is just that, a CAR.

91-93: Please give some explanation why a short half-life of signaling events speaks against direct TCR signal integration from multiple receptors and co-receptors. Also, please discuss if your system is sufficient to generate data to answer this question.

This sentence has now been modified to more correctly assert that our findings merely limit the window over which direct signal integration could occur, rather than ruling out completely.

108-110: Please discuss in more detail in which way this receptor more closely replicates TCR complex function than other CAR designs and what are limitations of it.

This sentence has been modified to make clear we are not trying to say our OptoCAR more closely resembles the native TCR, rather that it can elicit T cell function in a similar manner to the TCR and other CAR designs.

114-116: How does this 'most closely replicate the response to' TCR dissociation? Please give some reasons to choose intracellular or extracellular disruption for this question.

This sentence has been revised to make clear that intracellular dissociation minimises the potential disruption of cell conjugates if receptor/ligand interaction were the primary driver of conjugate formation.

125-127: Here you convincingly argue for studying single signaling pathways in isolation. However, please reconcile this statement with your ambition of studying multiple APC - T cell interactions (compare 92) and clarify these statements accordingly.

We have made clear that we wish to study the integration capabilities of the TCR signalling pathway rather than any combinatorial effect of other pathways, such as co-stimulation.

153-156: This section needs more explanations. Why does proximal signaling remain uninitiated? Does other signaling occur?

This sentence has been clarified to confirm it is OptoCAR proximal signaling that is not initiated because of the absence of dimerizer. Integrin and costimulatory receptors may signal but this is normally minimal without concomitant antigen receptor signalling.

167-170: This is a key control! It is unclear if this control was included for other experiments as well. E.g. for the same setup as 1 D and E and for other signaling pathways, this is of major importance to exclude light induced effects. In some figures different LOV2 domain mutants were used but not

clearly labeled. It would help clarity to always include the name of the LOV2 mutation in the respective figure. Please provide also non-normalized measurements for comparison and explain how you define 0 and 1?

As stated above for point 1, we have now included data using the same C450G/I539E variants for the other assays used in the manuscript. We have made sure that if any variant of LOV2 was used, this has been explicitly mentioned in the text and/or legend. We believe the reviewer is referring to Figure 1E when they mention defining 0 and 1, not figure 2. We have now included the raw plots used to extract the LOV2 kinetics in a supplementary figure, along with the C450G control of this experiment.

229: Figure 3B, 3E- did you perform negative controls with the LOV2 mutant for this experiment?

Yes, these controls have now been included in Figure EV2.

230-230: Please discuss this statement also in the context of non-normalized values. For the whole figure 3, from the figure alone it's difficult to understand what was done. Time of induction with the dimerizer and time of illumination should be more clearly labeled.

For figure 3, there is no normalisation as such. The raw histograms in parts A-C have simply been scaled to provide uniform area in each plot, which simply helps to more easily compare plots. This has no effect on the median ppERK values that are used to fit the data. There is no normalisation in panels D-F, just background correction. We have now added an introductory panel to figure 3, which hopefully provides more clarity for when the dimerizer is added the illumination times.

264-266: Due to all the normalizations in the data analysis in this figure I cannot make any comments on the relevance or accuracy of any of the conclusions.

The only normalisation performed on the datasets in figure 4C,D are background subtraction and scaling relative to the value at 180 minutes. Therefore, the profile of the graph is equivalent whether or not we do this processing. For figure 4F (previous version), we agree that it would be more helpful to also show the unnormalized data. This has now been added to the main figure, along with a schematic describing how the active fraction was quantified. We hope the reviewer is able to see the clear difference illumination makes and the obvious plateauing of the light-induced OFF after ~10 minutes.

300: reference missing

Added

301: reference missing

Added

316: Figure 5 All of the normalized graphs should also be compared to non normalized data and discussed in that context.

The only normalisation that has been to these datasets is background subtraction and scaling of datasets to the continuous pulse of 24 (B,D) or 6 hours (F,G). We have made this point clearer in the figure legend. Therefore, all the plots are readily comparable and there would be no difference in the profiles using the absolute values. Every single experiment performed using the OptoPlate included the 6-hour continuous control so all datasets could be fairly combined. It is likely that the dashed line in each panel does not help interpretation, and we have removed as it does not provide any more information.

352: Figure 5: x on the light intervals is not explained. E.g. 'light induced OFF' might be clearer. Also the relative lengths of on/off should be indicated. The blue is getting lighter to the right, giving the impression that light intensity was reduced, which is probably not intended, correct? The X label is unclear, shouldn't it be minutes instead of interval (min)? From the figure alone I cannot make out what it's supposed to show. The normalizations again make interpretations very difficult.

We have now included schematics for both the rationale of the experiment as well as showing what the 'pulse frequency modulation' means, in terms of light-induced Off interval and a pulse. We hope this now makes the experiment clearer. The faded blue on images the reviewer mentioned were intentionally stylistic to imply continuation of the pulses, but it is obviously not very helpful! We have now used an ellipsis to show that the pulse train continues. The x axis is the length of the light interval between signal pulses. This has now been more clearly labelled in figure.

375: Please explain why this is expected and provide a reference.

This sentence has been modified to suggest it as more of a conjecture for IL2.

382-386: Possibly except for IL2 all of the graphs in A show downregulation independent on illumination. Especially for FOS. Please explain your results in the context of this. Please also discuss other possibilities than mRNA half-life that could play a role.

In the text we mentioned the trajectory of mRNA expression in this context but weren't clear enough. It is expected in any gene-regulatory network that there are temporal phases of expression and decline. Our point we have hopefully made better in the manuscript is that these expression dynamics are still dependent on the state of the initiating receptor, and not hard-coded once gene expression starts.

RT-qPCR only measures the level of mRNA and not its sequestration or potential protein binding that might change gene expression. Therefore, any loss of detected mRNA must be due to its degradation.

399: Reference missing

Included

406: Please define 'rate of activation' and explain how your data supports negative feedback and why signaling pulses may be beneficial.

This panel has been removed as it does not sufficiently help the argument. It was merely the difference plot of the CD137 expression with time as a proxy for rate of expression. The text has also been changed here.

411-413: Doesn't this mean that continuous induction was better than pulsed after all? Please clarify. Different populations should be labeled in the figure, best with color codes. Extracting that information from the legends alone is difficult.

A pulsed 6-hour stimulus was found to be broadly equivalent to a continuous 24 hour equivalent. This implies that 4 times less input is needed, which may be beneficial for other reasons such as the inhibiting the induction of exhaustion. This point has been added to the text. The figure has also been modified to hopefully make it more clear what is occurring in this experiment.

419-421: What does this mean for a real TCRs or a CARs? What are limitations in these interpretations?

Sentence modified.

422: Is this statement pathway specific?

We have caveated this to focus the conclusion solely on the pathways we have investigated.

429-431: I don't see this claim justified by the data in the current form.

We have toned down this sentence.

432-433: Please discuss how your measurements justify to make claims regarding multiple APC interactions and limitations of them.

A sentence on the limitation of the CAR approach has been added to this paragraph.

455-457: This interpretation strikes me as somewhat simplistic. Could you discuss other factors that contribute to persistent cellular changes caused by signaling?

Perhaps the reviewer could make some suggestions, but we are not sure what persistent cell states could be induced by a transcription factor without a concomitant change in gene expression? We have made tried to make this point more clear.

482-484: Why do you argue that the OptoCAR is more equivalent to the TCR than other sorts of CARs and in what regards does it fall short?

This is not our intention, only to say that in higher ligand occupancy CARs, including our OptoCAR, behave similarly to the TCR. Text has been clarified.

592: The normalization is very unclear here, please show this plot without normalization for comparison. By eye there seems to be no very marked difference in FOS in the measured time, unlike what this plot shows and also unlike what panel E and figure 6 shows. Please discuss these discrepancies.

As stated above, we have included the unnormalized data in the main figure and have updated the figure legend to reflect this. Even for these plots, the decrease in the larger molecular weight version of FOS is clearly discernible.

605: What is the baseline here? The normalization should be explained better. Please include the non-normalized values for comparison and explain how you define 0 GFP in this plot.

The baseline is simply the auto-fluorescent background from non-activated cells, which has been subtracted from all values. This has been stated in the legend. As absolute GFP fluorescence from the flow cytometer is uncalibrated, whether the values are scaled to 1 or 1500 will make no difference to the figure. The raw plots are given in the EV figure so the reader can easily observe the >1 decade increase in both GFP and CD69 expression for comparison, which is the unnormalized data.

615: Please indicate in the figure the intervals for both light on and light off. From figure and legend it is unclear what the dotted line describes. The first pulse only and then light? At what time point is it shown?

As stated above, we have now hopefully improved the clarity of this figure, and also removed the dotted lines.

627: what is this 'defined period'? 3 hours?

This is correct and has been clarified in legend.

641: Please define 'control'

This has been clarified in legend and figure.

644-645: Please define what you mean by '24-hour range of signal pulses.' And compare to non normalized data.

There is no normalisation in this data, this was a text error. The data presented in panel is the actual CD137 surface expression values. Legend has been clarified.

885-886: I don't understand this sentence. Please clarify.

The figure now contains a schematic to more clearly explain this concept (based on another reviewer's suggestion) and we have now expanded this point in the methods. We hope this makes the point more clear.

913: Please also plot the total RNA levels relative to GAPDH and PGK1 as a comparison to your normalized data.

We have now included the cRQ values for the individual experiments in a supplementary figure to make it easier to see that all mRNA levels decay almost to baseline within 60 minutes.

927-928: This can be very misleading. Please show the graphs with the cRQ in the main figure. As of now, I cannot comment on any relevance or interpretation of the data, as normalization removes every scale of the effects. Please justify your choice of calculating mRNA half-life based on this normalized data.

Following on from the previous point, the cRQ values have now also been provided. We accept that the methods text implies a greater normalization of the data than was actually performed. The lowest value for all datasets is 1, the fold change over zero timepoint. We have simply subtracted 1 from the data to more easily show the datasets. As for the other assays, we have scaled the maximal value so that the kinetics can be more easily compared. Regardless of whether we do this or not, the fitting of the exponential decay rate would be unaffected by this data processing.

982-985: Please explain in more detail what you did here and how you interpret the results

This panel has now been removed from the manuscript.

989: Could you explain the purpose and your interpretation of this plot? Can you see the bands corresponding to protein of the upper panel or are they too faint?

Figure legend has been improved. The total protein normalization is a superior way to account for loading effects of samples on Western blots, in preference to GAPDH or tubulin, say. It is not expected to see one's protein of interest in the TPN blot.

1001: Could you please show data for baseline expression compared to WT cells?

We are not sure what is meant by this point. The baseline expression (vehicle control) is compared to WT cells, which is simply the OptoCAR-T cells maintained in the dark state. This sets the dynamic range of the assay, and why it has been included in the figure. We have added a point to the legend to make it clear the dark state provides the maximal (WT) output.

1020: This format gives much more relevant information. These plots really do look very different. Please discuss the uses of the normalized data for the analysis and justify how and when normalization is beneficial for data analysis and when it isn't.

The LHS panel in Figure S3 is the same data in the main figure, just all put in one light interval axis. Perhaps the reviewer is referring to the RHS panel with we do normalize the data to the minimum value? This was only to make the point the decay over ~10 minutes is equivalent, and we have now removed this panel as it easier enough to visualise the point made without it.

Thank you again for sending us your revised manuscript. We have now heard back from the three referees who were asked to evaluate the revised study. As you will see below, they are satisfied with the performed revisions and are supportive of publication. Reviewer #3 only lists a few remaining concerns, including points related to data availability (points #1 and #2). We would ask you to perform a minor revision to address these remaining issues as well as a few editorial issues listed below.

REFEREE REPORTS

Reviewer #1:

The revised manuscript addresses all my previous points satisfactorily. This is a high quality manuscript and I support publication.

Reviewer #2:

The authors have addressed all of my questions

Reviewer #3:

Thank you for your careful revision. Most of my questions have been addressed and the paper is now easier to follow. In my opinion, once some few remaining points are resolved, I view the manuscript a valuable addition to the current understanding of signaling dynamics ready for publication.

1. Major point: Please upload the annotated genbank files (as supplement or to genbank) of the whole sequence of all the plasmids used in this study. Giving the protein sequences only does not guarantee reproducibility. This is a requirement of the journal and the authors state that this has been done in the data accessibility section in the excel file but at the same time write 'This study includes no data deposited in external repositories.' In the manuscript.

2. 'the absolute values provide no additional meaning to the datasets.' I disagree on this point and would like to leave it to the editor to decide if these data sets provide value or not. Having the raw data in the supplements gives me the option to judge for myself and I can more easily draw conclusions about quantitative effects. Also, whoever wants to repeat the experiment will know what type of data to expect. As all the data exists, it seems to me to be a favorable benefit/work ratio to include it instead of discarding it, which in my opinion wastes potential resources for future studies for no apparent reason. Along this line, in the excel file, the authors indicate 'N/A' for the deposition of 'any datasets that are central and integral to the study'. Providing the underlying datasets for the figures helps independent data analysis and would enable machine readability, therefore I do think that this is a reasonable request.

3. My concerns about the data normalizations in the manuscript are independent on the wording 'normalized' or 'relative'. I appreciate the author's efforts to be more transparent about their data processing, but I would like to stress that for every interpretation of the results, it is essential to clearly label what the data is relative (or normalized) to. Indicating this directly in the figure will help the reader to understand the results much better. Instead of labeling the Y axis 'relative [A]' I would suggest to always provide the information what the value is relative to e.g. A/B.

For qPCR data normalizing to housekeeping genes is the standard, and should be labeled as such in the figure with units such as e.g. 10-x GAPDH. In the current format it is still not clear what the numbers on the Y axis actually stand for. Subtracting baseline expression may be applicable to zoom in on the effect of interest, but doing so requires providing the original datasets for reference as otherwise all quantitative information is lost. The alternative would be to start the Y axis at the baseline value to zoom in on the area of interest while still maintaining the quantitative information.

4. Throughout the introduction and results CAR and TCR is still largely used synonymously. A CAR and downstream signaling is not the same as a TCR and TCR signaling, as the authors point out well in the discussion. I think more precise wording in the other sections would help not blur this distinction.

The authors have made all requested editorial changes.

Thank you again for sending us your revised manuscript. We are now satisfied with the modifications made and I am pleased to inform you that your paper has been accepted for publication.

Corresponding Author Name: John Robert James

Manuscript Number: MSB-2020-10091